# *Xanthomonas citri* subsp. *citri* type III effector PthA4 directs the dynamical expression of a putative citrus carbohydrate-binding protein gene for canker formation

Xinyu Chen[1], Huasong Zou[2]*, Tao Zhuo[1], Wei Rou[1], Wei Wu[1], Xiaojing Fan[1]

[1]Plant Protection College, Fujian Agriculture and Forestry University, Fuzhou, China; [2]School of Life Sciences and Health, Huzhou College, Huzhou, China

*For correspondence:
zouhuasong@zjhzu.edu.cn

Competing interest: The authors declare that no competing interests exist.

**Abstract** *Xanthomonas citri* subsp. *citri* (*Xcc*), the causal agent of citrus canker, elicits canker symptoms in citrus plants because of the transcriptional activator-like (TAL) effector PthA4, which activates the expression of the citrus susceptibility gene *CsLOB1*. This study reports the regulation of the putative carbohydrate-binding protein gene *Cs9g12620* by PthA4-mediated induction of CsLOB1 during *Xcc* infection. We found that the transcription of *Cs9g12620* was induced by infection with *Xcc* in a PthA4-dependent manner. Even though it specifically bound to a putative TAL effector-binding element in the *Cs9g12620* promoter, PthA4 exerted a suppressive effect on the promoter activity. In contrast, CsLOB1 bound to the *Cs9g12620* promoter to activate its expression. The silencing of *CsLOB1* significantly reduced the level of expression of *Cs9g12620*, which demonstrated that *Cs9g12620* was directly regulated by CsLOB1. Intriguingly, PhtA4 interacted with CsLOB1 and exerted feedback control that suppressed the induction of expression of *Cs9g12620* by CsLOB1. Transient overexpression and gene silencing revealed that *Cs9g12620* was required for the optimal development of canker symptoms. These results support the hypothesis that the expression of *Cs9g12620* is dynamically directed by PthA4 for canker formation through the PthA4-mediated induction of CsLOB1.

## eLife assessment

This **valuable** study provides new insight into potential subtle dynamics in effector biology. The data presented generally support the claims, but in some cases, significant controls are missing and so the overall work is currently **incomplete**. If the limitations can be addressed, this work should be of broad relevance for biologists interested in molecular plant-microbe interactions.

## Introduction

Citrus canker is a severe bacterial disease that impacts the production of citrus because it affects most commercial cultivars of citrus in tropical and subtropical areas. The early symptoms on leaves, stems, and fruits are characterized by tiny, slightly raised blister-like lesions with water-soaked margins, which then develop into necrotic, raised lesions (*Brunings and Gabriel, 2003*). In addition to the severe defoliation and premature fruit dehiscence that usually occur on heavily infected trees, the blemished fruits also have significantly reduced commercial value. The citrus canker is associated with five different bacterial pathotypes, including three pathotypes of *Xanthomonas citri* subsp. *citri* (*Xcc*) (A,

A*, and Aʷ) and two pathotypes of *X. fuscans* subsp. *aurantifolii* (B and C) (*Gottwald et al., 2002*; *Sun et al., 2004*). The *Xcc* A-type canker is the most prevalent and affects a wide range of hosts, including *Citrus* spp. and many closely related rutaceous plants (*Graham et al., 2004*).

*Xcc* PthA4 is a transcription activator-like (TAL) effector that is delivered into host plants by a type III secretion system (T3SS). The *Xcc* strains defective in the *pthA4* gene cannot cause canker symptoms on their normal hosts (*Al-Saadi et al., 2007*; *Yan and Wang, 2012*). Uniquely, *pthA4* alone induces weak canker phenotypes in citrus leaves (*Al-Saadi et al., 2007*). The heterologous expression of T3SS and *pthA4* homologous genes in *Escherichia coli* elicits similar canker symptoms in citrus but not in tobacco (*Nicotiana benthianum*), soybean (*Glycine max*), or cotton (*Gossypium hirsutum*) (*Kanamori and Tsuyumu, 1998*). Variants of the PthA4 effector are widely present across all five bacterial subspecies that cause citrus canker disease (*Swarup et al., 1992*; *Al-Saadi et al., 2007*; *Jalan et al., 2013*). PthA4 acts as a major virulence determinant for the formation of cankers when translocated into the nuclei of its citrus host (*Yang and Gabriel, 1995*). The superhelical structure of this TAL effector recognizes host sequences through its repeat variable di-residue (RVD), which specifically recognizes the effector-binding element (EBE) in the gene promoter of host plants, and thus, activates the expression of host target gene (*Deng et al., 2012*). Notably, PthA4 acts in a protein–protein interaction manner during infection with *Xcc* (*Domingues et al., 2010*). In addition to the proteins associated with DNA repair (*Domingues et al., 2010*; *de Souza et al., 2012*), PthA4 specifically interacts with citrus CsMAF1 and releases the suppressive effect on RNA Polymerase III to increase the transcription of tRNA and cell proliferation (*Soprano et al., 2013*). Thus, *Xcc* PthA4 functions through versatile mechanisms and modulates target genes not only at the transcriptional level but also at the protein level.

PthA4 activates the transcription of the *CsLOB1* susceptibility gene in host citrus plants owing to the TAL effector DNA-binding principles and specificity (*Moscou and Bogdanove, 2009*; *Boch et al., 2009*; *Hu et al., 2014*; *Li et al., 2014*). Remarkably, gene editing of the coding region or promoter sequence of *CsLOB1* has been used to promote the resistance of citrus to infection with *Xcc* (*Jia et al., 2016*; *Peng et al., 2017*). *CsLOB1* belongs to the lateral organ boundary domain (LBD) gene family, which encodes a set of plant-specific transcription factors (TFs) that contain cysteine-rich DNA-binding motifs (*Husbands et al., 2007*). Plant LBD genes are involved in the regulation of plant growth, as well as the production of anthocyanins and nitrogen metabolism, and also respond to both hormonal and environmental stimuli (*Majer and Hochholdinger, 2011*). There are as many as 36 LBD proteins in citrus plants. CsLOB2 and CsLOB3 show the same involvement in inducing the formation of pustules as CsLOB1 when they are induced by *Xcc*, which expresses an artificial TAL effector (*Zhang et al., 2017*). In this case, many differentially expressed citrus genes have been identified, including those related to cell wall metabolism. However, the CsLOB2 and CsLOB3 promoters do not contain specific binding elements for PthA4; thus, neither can be induced by PthA4 during infection by *Xcc* (*Zhang et al., 2017*).

Since *CsLOB1* is a TF, elucidating its downstream target genes is particularly important to understand the molecular events associated with the development of pustule symptoms (*Hu et al., 2014*; *Li et al., 2014*). PthA4 and CsLOB1 form a regulatory cascade that activate the citrus genes during *Xcc* infection (*Duan et al., 2018*). Our previous research showed that the ectopic expression of the TAL gene *avrXa7* in *Xcc* suppressed the development of cankers but induced a yellowing phenotype around the inoculation site (*Sun et al., 2018*). Moreover, 138 upregulated citrus plant genes were identified to respond to wildtype (WT) *Xcc* infection but were relatively downregulated in leaves inoculated with *Xcc* 29-1 that expresses *avrXa7*. This study aimed to characterize the target genes of the citrus hosts that are directly regulated by the PthA4-mediated induction of CsLOB1 during *Xcc* infection, as well as its potential role in canker development.

## Results

### The expression of *Cs9g12620* depends on *pthA4* during *Xcc* infection

Our previous study on a transcriptome analysis reported that 138 genes were upregulated in citrus plants infected with the WT *Xcc* and downregulated in plants inoculated with *Xcc* that expressed *avrXa7*. Those genes were assumed to be related to the major *Xcc* virulence effector PthA4. PthA4 has 17.5 RVDs in its central domain that correspond to an EBE of 18 nucleotides of the target gene

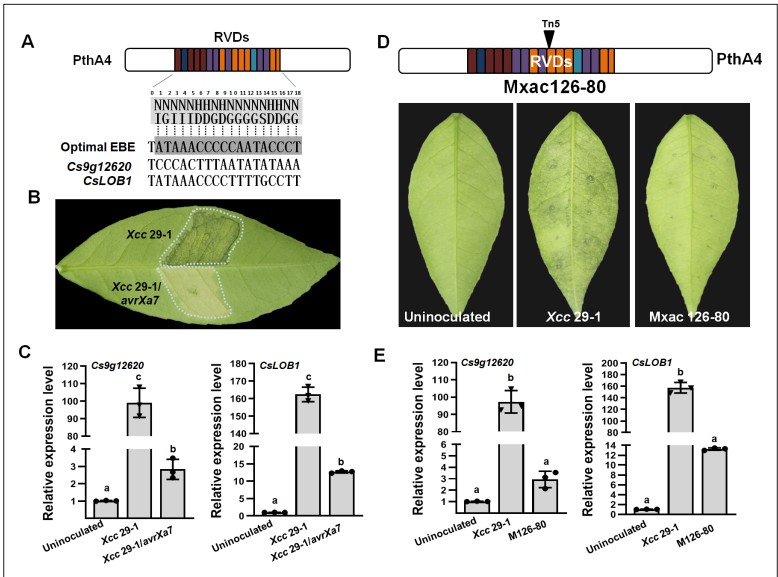

**Figure 1.** The expression of *Cs9g12620* depends on PthA4 during *Xcc* infection. (**A**) Predicted PthA4-binding elements of the *Citrus sinensis* genes upregulated in response to *Xcc* infection. The optimal effector-binding element (EBE) that corresponded to the PthA4 repeat variable di-residues (RVDs) were predicted based on the transcriptional activator-like (TAL) effector–DNA-binding code. Putative PthA4 EBEs in promoters of *Cs9g12620* and *CsLOB1* are shown below the optimal EBE of PthA4. (**B**) The disease symptoms of *Xcc* 29-1/avrXa7 on *C. sinensis* leaves. The cell suspension ($10^8$ CFU/ml) was infiltrated into plant leaves, and the phenotype was recorded at 5 dpi. All the experiments were repeated three times with similar results. (**C**) A qRT-PCR analysis of the expression of *Cs9g12620* and *CsLOB1* in plants inoculated with *Xcc* that expressed *avrXa7*. The level of expression of each gene in the uninoculated samples was set as 1, and the levels in other samples were calculated relative to those baseline values. Values are the mean results from three biological replicates and are the means ± SD (ANOVA, p < 0.01). The experiment was repeated three times. (**D**) The phenotype elicited by the *pthA4* mutant Mxac126-80. The analyses were performed as described in **B**. (**E**) qRT-PCR analysis of the transcript levels of *Cs9g12620* and *CsLOB1* plants inoculated with *Xcc* 29-1 and Mxac126-80. The analyses were the same as those conducted in **C**. ANOVA, analysis of variance; dpi, days post-inoculation; qRT-PCR, real-time quantitative reverse transcription PCR; SD, standard deviation; *Xcc, Xanthomonas citri* subsp. *citri*.

The online version of this article includes the following source data and figure supplement(s) for figure 1:

**Source data 1.** Excel file containing qRT raw data in *Figure 1C, E*.

**Source data 2.** Original image for the disease symptoms analysis in *Figure 1B* (*Xcc* 29-1, *Xcc* 29-1/avrXa7).

**Source data 3.** PDF containing *Figure 1B* and original scans of the disease symptoms analysis (*Xcc* 29-1, *Xcc* 29-1/avrXa7) with highlighted bands.

**Source data 4.** Original image for the disease symptoms analysis in *Figure 1D* (uninoculated).

**Source data 5.** Original image for the disease symptoms analysis in *Figure 1D* (*Xcc* 29-1).

**Source data 6.** Original image for the disease symptoms analysis in *Figure 1D* (Mxac126-80).

**Figure supplement 1.** The location of the PthA4 effector-binding element (EBE) in the promoter of *Cs9g12620* in *Citrus sinensis*.

promoter. Based on the TAL effector–DNA-binding code, PthA4 can bind an optimal EBE of the 19 bp oligonucleotide -TATAAACCCCCAATACCCT-, in which the first nucleotide 'T' at 0 position is strictly conserved (*Figure 1A*, *Figure 1—figure supplement 1*). For each of the 138 genes, we downloaded and analyzed the 2000 bp DNA sequence upstream of the translation start site from the public citrus genome database (https://www.citrusgenomedb.org/). Possible PthA4 EBEs were identified in the promoter region of the two genes *Cs9g12620* and *CsLOB1* (*Cs7g27640.1*) (*Figure 1A*).

The transcript levels of *Cs9g12620* and *CsLOB1* were studied to determine whether they were associated with PthA4. In view of the transcriptome analysis, their transcript levels were first studied in sweet orange (*Citrus sinensis*) plants inoculated with the WT *Xcc* 29-1 and *Xcc* 29-1 that expressed *avrXa7*. The inoculation of *Xcc* 29-1 that expressed *avrXa7* did not cause canker symptoms and

displayed a yellow phenotype (*Figure 1B*). In comparison with the uninoculated control, the level of expression of *Cs9g12620* was induced remarkably by inoculation with the WT *Xcc* 29-1 but not *Xcc* 29-1 that expressed *avrXa7* (*Figure 1C*). Their transcript levels were then studied in *C. sinensis* plants inoculated with the *pthA4* Tn5 insertion mutant Mxac126-80, which was impaired in its ability to cause canker symptoms (*Figure 1D*). In comparison with the plants inoculated with the WT *Xcc* 29-1, *Cs9g12620*, and *CsLOB1* were downregulated in the plants infected with Mxac126-80 (*Figure 1E*). These findings demonstrated that *Cs9g12620* was activated by *Xcc* infection in a PthA4-dependent manner.

## *Cs9g12620* shows different profiles of expression from its homolog *Cs9g12650* owing to the genetic diversity in promoter region

A BLAST search was performed in the citrus genome database using the *Cs9g12620* sequence. A highly conserved homolog *Cs9g12650* was found closely downstream of *Cs9g12620* in chromosome 9 of the *C. sinensis* chromosome (*Figure 2A*). Their amino acid sequences are 86% homologous. The existence of *Cs9g12620* and *Cs9g12650* in the *C. sinensis* chromosome was verified by PCR amplification with two primer sets anchored to the two genes, respectively (*Figure 2B*). Their transcription was then assessed by semi-quantitative reverse transcription PCR (RT-PCR) analysis. The results showed that *Cs9g12620* was expressed at a high transcript level, while no transcripts of *Cs9g12650* were found, even in the plants inoculated with the WT *Xcc* 29-1 (*Figure 2C*).

A sequence analysis of the *Cs9g12620* and *Cs9g12650* promoter regions revealed that are 71% homologous. A prediction of the promoter in Softberry (http://linux1.softberry.com/berry.phtml) identified the core structure region from the same location in both the *Cs9g12620* and *Cs9g12650* promoters (*Figure 2—figure supplement 1*). It should be noted that the predicted PthA4 EBE was located in the promoter core structure. A sequence alignment indicated that the last 22 nucleotides at the 3′ terminus in *Cs9g12620* promoter had been lost genetically in the *Cs9g12650* promoter (*Figure 2—figure supplement 1*). Fragments of 463 bp in the *Cs9g12620* promoter and 460 bp in the *Cs9g12650* promoter that composed the core structure were fused with a luciferase reporter. The transient overexpression of the promoter luciferase fusion indicated that the *Cs9g12650* promoter was unable to drive the expression of luciferase in *Nicotiana benthamiana* (*Figure 2D*). This demonstrated that the *Cs9g12650* promoter region lacked promoter activity. To confirm the effect of 22 nucleotides on promoter activity, we subsequently created a *Cs9g12620* promoter mutant with a deletion of 22 nucleotides at the 3′ terminus. The mutant completely lost its ability to drive the expression of luciferase, suggesting that the 22 nucleotides are essential for activity of the promoter (*Figure 2E*).

## PthA4 suppresses *Cs9g12620* promoter activity independent of the binding action

The PthA4-dependent induction of *Cs9g12620* during *Xcc* infection prompted us to determine whether it is directly regulated by PthA4. The 463 bp *Cs9g12620* promoter was used in yeast one-hybrid (Y1H) assays. The EGY48 yeast co-transformed with pGBKT7-pthA4 and pG221 showed weak β-galactosidase activity, which demonstrated that PthA4 functions as a transcriptional activator. The co-transformation of pGBKT7-pthA4 and pG221-P*Cs9g12620* resulted in deep blue, suggesting that PthA4 binds the *Cs9g12620* promoter. The enhanced β-galactosidase activity was quantified by the Miller method (*Figure 3A*). In an electrophoretic mobility shift assay (EMSA), the application of 0.117, 0.469, 1.875, 7.5, 30, and 120 µg of GST-PthA4 proteins partially hindered the mobility of 25 ng *Cs9g12620* promoter DNA. The application of 7.5, 30, and 120 µg GST-PthA4 proteins completely retarded the mobility of 25 ng promoter DNA (*Figure 3B*). The negative control glutathione-*S* transferase (GST) tag was unable to bind the *Cs9g12620* promoter DNA.

To examine whether the activity was affected by PthA4, the *Cs9g12620* promoter was cloned into the binary vector pCAMBIA1381 fused to a *gusA* reporter gene. Co-transformation with *pthA4* did not induce the *Cs9g12620* promoter activity. Histochemical staining revealed that the β-glucuronidase (GUS) activity in the leaves that co-expressed *pthA4* suppressed the promoter activity (*Figure 3C*). For further verification, the levels of expression of *gusA* and *pthA4* were evaluated by real-time quantitative reverse transcription PCR (qRT-PCR). Compared with the expression of the promoter GUS fusion alone or in combination with the empty vector pHB control, the transcript level of *gusA* was reduced when the *Cs9g12620* promoter GUS fusion was co-transformed with *pthA4* (*Figure 3C*).

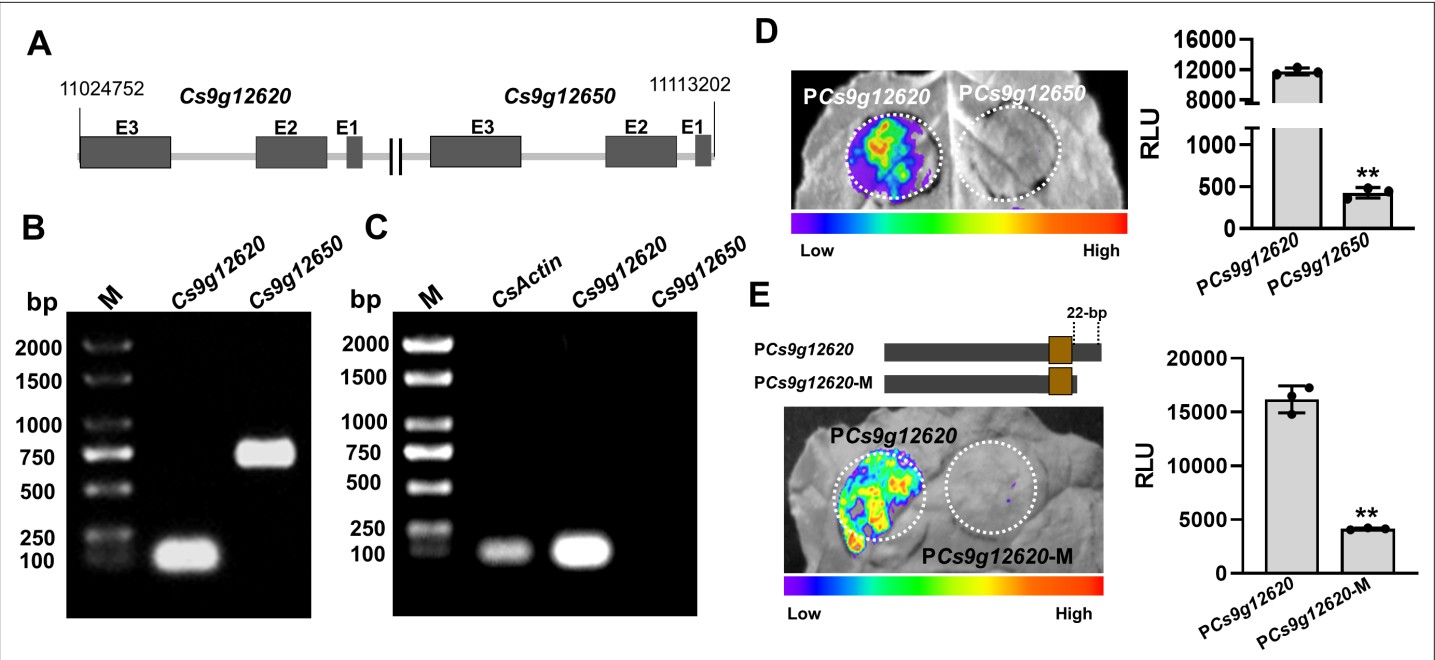

**Figure 2.** Cs9g12620 and Cs9g12650 show different profiles of expression owing to the genetic variation in promoter. (**A**) The location of *Cs9g12620* and *Cs9g12650* in the *Citrus sinensis* chromosome. The location was revealed by a BLAST search of the Citrus Genome Database (https://www.citrusgenomedb.org/). (**B**) Verification of the existence of *Cs9g12620* and *Cs9g12650* in the *C. sinensis* genome by PCR amplification. The specific primers for either gene were used for qRT-PCR analysis, which are shown in *Supplementary file 1b*. M, DL2000 DNA marker. (**C**) Semi-quantitative RT-PCR detection of the transcription of *Cs9g12620* and *Cs9g12650. CsActin* was used as the internal control gene. A total of 2 µg of total RNA extracted from *C. sinensis* was used to synthesize the first single-stranded cDNA. The specific primers for each gene were the same as those of the qRT-PCR analysis and are shown in *Supplementary file 1b*. M, DL2000 DNA marker. (**D**) Luciferase assays of *Cs9g12620* and *Cs9g12650* promoter activity. The *Cs9g12620* and *Cs9g12650* promoter luciferase fusions were transiently expressed in the *Nicotiana benthamiana* leaves. Luciferase activity was measured with a CCD imaging system at 2 days post-agroinfiltration. The image on the right shows the quantification of luciferase signal using a microplate luminescence reader. Values are the means ± SD (*n* = 3 biological replicates; Student's *t*-test, **p < 0.01). (**E**) Luciferase assays of *Cs9g12620* and *Cs9g12620*-M with the deletion of 22 bp at the 3'-terminus. The *Cs9g12620* and *Cs9g12620*-M promoter luciferase fusions were transiently expressed in the *N. benthamiana* leaves. Luciferase activity was measured with a CCD imaging system at 2 days post-agroinfiltration. The luciferase signal was quantified as described in **D**. CCD, charge-coupling device; qRT-PCR, real-time quantitative reverse transcription PCR; RT-PCR, reverse transcription PCR; SD, standard deviation.

The online version of this article includes the following source data and figure supplement(s) for figure 2:

**Figure supplement 1.** Alignment of the promoter regions of *Cs9g12620* and *Cs9g12650* in *Citrus sinensis*.

**Source data 1.** Excel file containing RLU raw data in *Figure 2D, E*.

**Source data 2.** Original file for the PCR amplification analysis in *Figure 2B* (*Cs9g12620* and *Cs9g12650*).

**Source data 3.** PDF containing *Figure 2B* and original scans of the PCR amplification analysis (*Cs9g12620* and *Cs9g12650*) with highlighted bands.

**Source data 4.** Original file for the semi-quantitative reverse transcription PCR (RT-PCR) analysis in *Figure 2C* (*CsActin*, *Cs9g12620*, and *Cs9g12650*).

**Source data 5.** PDF containing *Figure 2C* and original scans of the semi-quantitative reverse transcription PCR (RT-PCR) analysis (*CsActin*, *Cs9g12620*, and *Cs9g12650*) with highlighted bands.

**Source data 6.** Original file for the luciferase assays analysis in *Figure 2D* (*Cs9g12620* and *Cs9g12650* promoter).

**Source data 7.** PDF containing *Figure 2D* and original scans of the luciferase assays analysis (*Cs9g12620* and *Cs9g12650* promoter) with highlighted bands.

**Source data 8.** Original file for the luciferase assays analysis in *Figure 2E* (*Cs9g12620* and *Cs9g12620*-M promoter).

**Source data 9.** PDF containing *Figure 2E* and original scans of the luciferase assays analysis (*Cs9g12620* and *Cs9g12620*-M promoter) with highlighted bands.

In the EMSA analysis that utilized a synthetic PthA4 EBE (-TCCCACTTTAATATATAAA-) as a probe, 6, 12, and 24 µg of GST-PthA4 protein partially hindered the mobility of oligonucleotide (*Figure 3D*). This demonstrated that PthA4 truly bound to the predicted EBE located at the promoter core structure (*Figure 2—figure supplement 1*). The first base pair 'T' at 0 position in the sequence was then

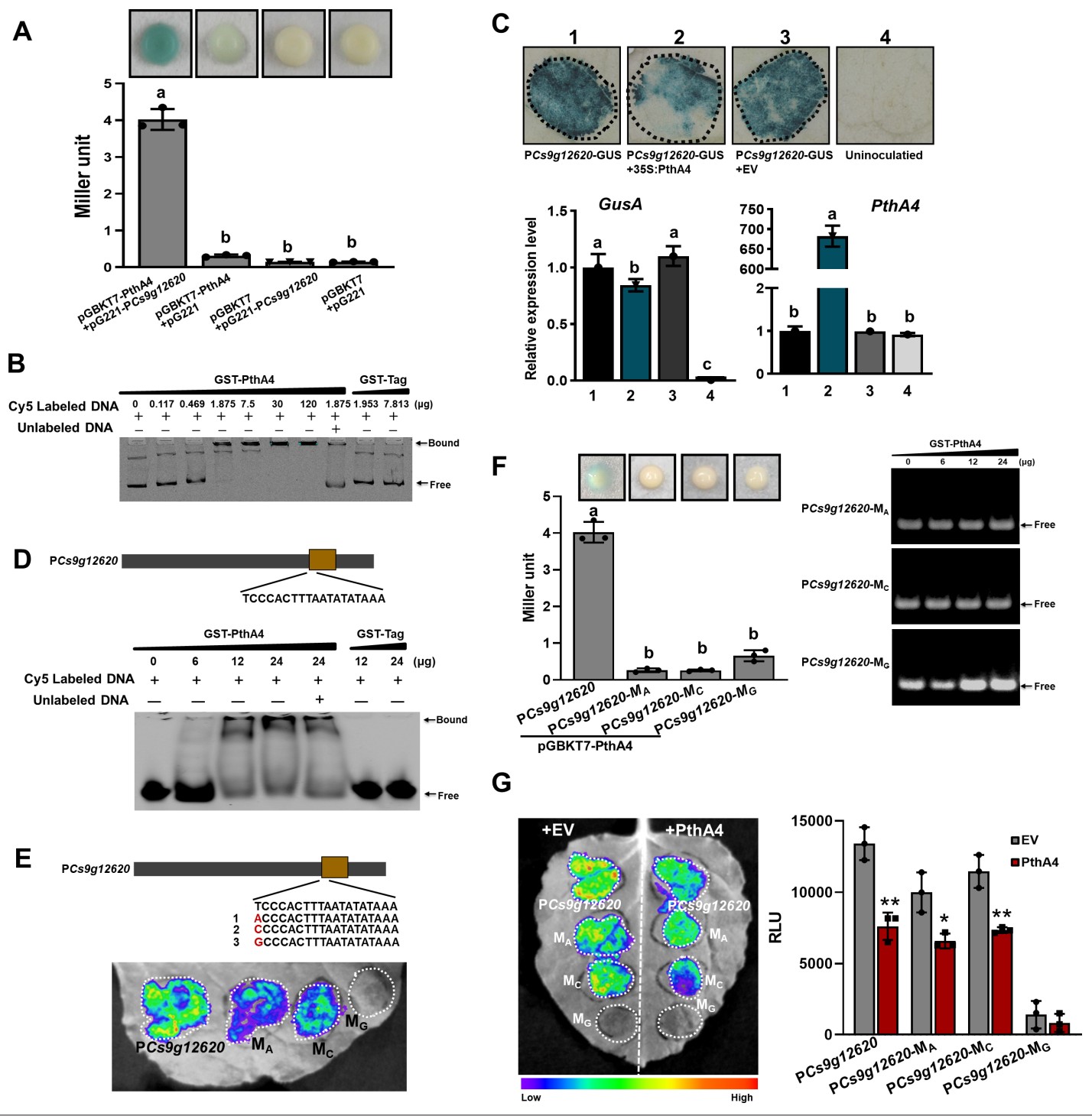

**Figure 3.** PthA4 binds to and suppresses the *Cs9g12620* promoter. (**A**) A yeast one-hybrid assay shows that PthA4 interacts with the *Cs9g12620* promoter. pGBKT7-pthA4 and pG221-P*Cs9g12620* were co-transformed into the yeast strain EGY48 and screened on synthetic dropout SD/-Ura and SD/-Ura/-Trp media. The transformed cells were dissociated by repeated freeze–thaw treatment and then used for β-galactosidase assays on filters soaked with Z buffer that contained 20 µg/ml X-gal. The pGBKT7 and pG221 vectors were used as negative controls by co-transformation with the corresponding constructs. The columns below show the quantitative assay of β-galactosidase by the Miller method. Values are the means ± SD (*n* = 3 biological replicates). Columns labeled with different letters indicate significant difference among means (ANOVA, p < 0.01). (**B**) An electrophoretic mobility shift assay (EMSA) shows that PthA4 binds to the *Cs9g12620* promoter. Purified GST-PthA4 was incubated with 25 ng of promoter DNA in gel shift binding buffer at 28°C for 30 min and then analyzed in 6% non-denaturing PAGE. A GST tag was used as a negative control. (**C**) The overexpression of PthA4 suppressed the *Cs9g12620* promoter. PthA4 was transiently co-expressed with a *Cs9g12620* promoter GUS fusion in *Nicotiana benthamiana*.

*Figure 3 continued on next page*

*Figure 3 continued*

The promoter-driven GUS activity was assayed at 2 days post-agroinfiltration. The upper images show histochemical staining of the *Cs9g12620* promoter-driven GUS activity. The bar chart below shows the qRT-PCR analysis of the transcript levels of *gusA* and *pthA4*. The level of expression of each gene in the samples that expressed the promoter alone was set as 1, and the levels of the other samples were calculated relative to that. Values are the mean results from three biological replicates and are the means ± SD. Columns labeled with different letters indicate significant difference among means (ANOVA, $p < 0.01$). (**D**) EMSA shows that PthA4 binds to the 19 bp predicted binding site (effector-binding element, EBE) in the *Cs9g12620* promoter. The purified GST-PthA4 was incubated with 25 ng of synthetic 19 bp EBE DNA fragment in gel shift binding buffer at 28°C for 30 min and then analyzed in 6% non-denaturing PAGE. (**E**) Luciferase assays of the activity of *Cs9g12620* promoter with site-directed mutation in PthA4 EBE. The first nucleotide acid 'T' in EBE was mutated into A, C, or G, which generated P*Cs9g12620*-$M_A$, P*Cs9g12620*-$M_C$, and P*Cs9g12620*-$M_G$, respectively. The mutants were fused with luciferase and transiently overexpressed in *N. benthamiana* leaves. The luciferase activity was measured with a CCD imaging system at 2 days post-agroinfiltration. (**F**) PthA4 did not bind to P*Cs9g12620*-$M_A$, P*Cs9g12620*-$M_C$, and P*Cs9g12620*-$M_G$. The left image represents a yeast one-hybrid assay that did not show an interaction of PthA4 between P*Cs9g12620*-$M_A$, P*Cs9g12620*-$M_C$, and P*Cs9g12620*-$M_G$. The experiments were conducted in the same manner as those in **A**. Columns labeled with different letters indicate significant difference among means (ANOVA, $p < 0.01$). The right image represents an EMSA that shows that PthA4 does not bind to the promoter mutants. Purified GST-PthA4 was incubated with 25 ng of promoter DNA in gel shift binding buffer at 28°C for 30 min and then analyzed in 6% non-denaturing PAGE. (**G**) Luciferase assays showing the suppression of PthA4 on P*Cs9g12620*-$M_A$, P*Cs9g12620*-$M_C$, and P*Cs9g12620*-$M_G$. The promoter luciferase fusions were co-expressed with PthA4 in *N. benthamiana*. Co-expression with the empty binary vector pHB was used as the control. The analysis was the same as that described in **E**. The image on the right shows the quantification of luciferase signal using a microplate luminescence reader. Values are the means ± SD ($n = 3$ biological replicates). ANOVA, analysis of variance; CCD, charge-coupling device; GST, glutathione-*S* transferase; PAGE, polyacrylamide gel electrophoresis; qRT-PCR, real-time quantitative reverse transcription PCR; SD, standard deviation. $^*p < 0.05$. $^{**}p < 0.01$ (Student's *t*-test).

The online version of this article includes the following source data for figure 3:

**Source data 1.** Excel file containing LacZ, qRT and RLU raw data in ***Figure 3A, C, F, G***.

**Source data 2.** Original file for the yeast one-hybrid assay in ***Figure 3A*** (pGBKT7-PthA4+pG221-P*Cs9g12620*, pGBKT7+pG221-P*Cs9g12620*, and pGBKT7+pG221).

**Source data 3.** Original file for the yeast one-hybrid assay in ***Figure 3A*** (pGBKT7-PthA4+pG221).

**Source data 4.** PDF containing ***Figure 3A*** and original scans of the yeast one-hybrid assay (pGBKT7-PthA4+pG221-P*Cs9g12620*, pGBKT7+pG221-P*Cs9g12620*, pGBKT7+pG221, and pGBKT7-PthA4+pG221) with highlighted bands.

**Source data 5.** Original file for the electrophoretic mobility shift assay in ***Figure 3B*** (GST-PthA4+*PCs9g12620*).

**Source data 6.** PDF containing ***Figure 3B*** and original scans of the electrophoretic mobility shift assay (GST-PthA4+*PCs9g12620*) with highlighted bands.

**Source data 7.** Original file for histochemical staining of the *Cs9g12620* promoter-driven GUS activity in ***Figure 3C*** (P*Cs9g12620*-GUS, P*Cs9g12620*-GUS+35S:PthA4, P*Cs9g12620*-GUS+EV, and uninoculatied).

**Source data 8.** PDF containing ***Figure 3C*** and original scans of the histochemical staining of the *Cs9g12620* promoter-driven GUS activity assay (P*Cs9g12620*-GUS, P*Cs9g12620*-GUS+35S:PthA4, P*Cs9g12620*-GUS+EV, and uninoculatied) with highlighted bands.

**Source data 9.** Original file for the electrophoretic mobility shift assay in ***Figure 3D*** (GST-PthA4+19 bp effector-binding element [EBE] P*Cs9g12620*).

**Source data 10.** PDF containing ***Figure 3D*** and original scans of the electrophoretic mobility shift assay (GST-PthA4+19 bp effector-binding element [EBE] P*Cs9g12620*) with highlighted bands.

**Source data 11.** Original file for the luciferase assays in ***Figure 3E*** (P*Cs9g12620*, P*Cs9g12620*-$M_A$, P*Cs9g12620*-$M_C$, and P*Cs9g12620*-$M_G$).

**Source data 12.** PDF containing ***Figure 3E*** and original scans of the luciferase assays (P*Cs9g12620*, P*Cs9g12620*-$M_A$, P*Cs9g12620*-$M_C$, and P*Cs9g12620*-$M_G$) with highlighted bands.

**Source data 13.** Original file for the yeast one-hybrid assay in ***Figure 3F*** (pGBKT7-PthA4+pG221-P*Cs9g12620*).

**Source data 14.** Original file for the yeast one-hybrid assay in ***Figure 3F*** (pGBKT7-PthA4+pG221-P*Cs9g12620*-$M_A$, pGBKT7-PthA4+pG221-P*Cs9g12620*-$M_C$, and pGBKT7-PthA4+pG221-P*Cs9g12620*-$M_G$).

**Source data 15.** PDF containing ***Figure 3F*** and original scans of the yeast one-hybrid assay (pGBKT7-PthA4+pG221-P*Cs9g12620*, pGBKT7-PthA4+pG221-P*Cs9g12620*-$M_A$, pGBKT7-PthA4+pG221-P*Cs9g12620*-$M_C$, and pGBKT7-PthA4+pG221-P*Cs9g12620*-$M_G$) with highlighted bands.

**Source data 16.** Original file for the electrophoretic mobility shift assay in ***Figure 3F*** (GST-PthA4+P*Cs9g12620*-$M_A$ and GST-PthA4+P*Cs9g12620*-$M_C$).

**Source data 17.** Original file for the electrophoretic mobility shift assay in ***Figure 3F*** (GST-PthA4+P*Cs9 g12620*-$M_G$).

**Source data 18.** PDF containing ***Figure 3F*** and original scans of the electrophoretic mobility shift assay (GST-PthA4+P*Cs9g12620*-$M_A$, GST-PthA4+P*Cs9g12620*-$M_C$, and GST-PthA4+P*Cs9 g12620*-$M_G$) with highlighted bands.

**Source data 19.** Original file for the luciferase assays in ***Figure 3G*** (P*Cs9g12620*, P*Cs9g12620*-$M_A$, P*Cs9g12620*-$M_C$, and P*Cs9g12620*-$M_G$ co-expression with PthA4 and EV, respectively).

**Source data 20.** PDF containing ***Figure 3G*** and original scans of the luciferase assays (P*Cs9g12620*, P*Cs9g12620*-$M_A$, P*Cs9g12620*-$M_C$, and P*Cs9g12620*-$M_G$ co-expression with PthA4 and EV, respectively) with highlighted bands.

mutated into 'A', 'C', and 'G' in the *Cs9g12620* promoter, which generated the point mutations P*Cs9g12620*-$M_A$, P*Cs9g12620*-$M_C$, and P*Cs9g12620*-$M_G$, respectively. The activities of P*Cs9g12620*-$M_A$ and P*Cs9g12620*-$M_C$ were reduced compared with the WT P*Cs9g12620*. In contrast, the mutation 'T' to 'G' resulted in a complete loss of promoter activity (*Figure 3E*). The Y1H and EMSA assays demonstrated that PthA4 could not bind P*Cs9g12620*-$M_A$, P*Cs9g12620*-$M_C$, or P*Cs9g12620*-$M_G$ (*Figure 3F*). However, the co-transformation of PthA4 with P*Cs9g12620*-$M_A$, P*Cs9g12620*-$M_C$, and P*Cs9g12620*-$M_G$ retained the ability to suppress the promoter activity (*Figure 3G*).

## CsLOB1 binds multiple sites to induce the *Cs9g12620* promoter

CsLOB1 showed transcription activity since β-galactosidase activity was found when it was co-transformed with pG221 into yeast EGY48. The β-galactosidase activity was remarkably enhanced when CsLOB1 was co-transformed with pG221, which harbored P*Cs9g12620* (*Figure 4A*). The binding of CsLOB1 to P*Cs9g12620* was further verified by an EMSA analysis (*Figure 4B*). Most importantly, the *Cs9g12620* promoter activity was significantly induced when CsLOB1 was co-transformed with the P*Cs9g12620*-GUS fusion (*Figure 4C*).

Two putative CsLOB1-binding sites (-CGGC-) were found in the *Cs9g12620* promoter and located at the −268 and −151 positions (designated LB1 and LB2, respectively) (*Figure 4D*, *Figure 2—figure supplement 1*). To examine the role of the two CsLOB1-binding sites, the two binding sites LB1 and LB2 were deleted to generate the mutants P*Cs9g12620*-$M_{LB1}$, P*Cs9g12620*-$M_{LB2}$, and *Cs9g12620*-$M_{LB1/2}$. Deletion of the LB1 or LB2 site did not block the binding of CsLOB1 to P*Cs9g12620* in yeast (*Figure 4D*). In the EMSA analysis, CsLOB1 retarded the mobility of P*Cs9g12620*-$M_{LB1}$, P*Cs9g12620*-$M_{LB2}$, and P*Cs9g12620*-$M_{LB1/2}$ DNA (*Figure 4E*). Furthermore, the activities of these three promoter mutants were induced by CsLOB1 (*Figure 4F*). These results demonstrated that CsLOB1 binds to multiple sites in the *Cs9g12620* promoter to induce promoter activity. Citrus tristeza virus (CTV)-induced gene silencing was conducted in *C. sinensis* plants, which generated *CsLOB1*-silenced plants. The efficiency of infection was evaluated by analyses of the expression of *P23* gene harbored in the CTV vector. In comparison with the empty CTV-infected plants, the transcript level of *Cs9g12620* was significantly reduced in the *CsLOB1*-silenced plants (*Figure 4G*).

## PthA4 dynamically regulates the expression of *Cs9g12620* during *Xcc* infection

To examine the exact effect of PthA4 on the *Cs9g12620* promoter during *Xcc* infection, the *CsLOB1* promoter GUS fusion (P*CsLOB1*-GUS) was first co-transformed with PthA4. A comparison with P*CsLOB1*-GUS expressed alone showed that the expression of PthA4 induced the P*CsLOB1* promoter activity (*Figure 5A*). The co-expression of CsLOB1 with PthA4 had no effect on the PthA4-induced CsLOB1 promoter activity. This demonstrated that CsLOB1 is directly regulated by PthA4. When PthA4 and CsLOB1 were co-expressed with the *Cs9g12620* promoter, the expression of PthA4 strikingly suppressed the CsLOB1-induced *Cs9g12620* promoter activity (*Figure 5B*). To verify the suppressive effect by PthA4, *A. tumefaciens* GV3101 that expressed PthA4-FLAG was arranged at diverse concentrations of $OD_{600}$ values of 0.05, 0.1, 0.2, and 0.4. The results showed that the efficiency of suppression correlated with the high level of expression of PthA4-FLAG (*Figure 5C*).

To study the regulatory pattern of *pthA4* on the expression of *Cs9g12620* during canker development, the levels of expression of *pthA4*, *CsLOB1*, and *Cs9g12620* were quantitatively assayed in plants inoculated with the WT *Xcc* 29-1. Even though the levels of expression of *pthA4*, *CsLOB1*, and *Cs9g12620* were enhanced at 5 and 10 days post-inoculation (dpi) compared with 0 dpi, their pattern of expression differed between 5 and 10 dpi. *pthA4* and *CsLOB1* showed the highest levels of expression at 10 dpi. In contrast, the level of expression of *Cs9g12620* at 10 dpi was lower than that at 5 dpi (*Figure 5D*). To verify this special pattern of expression, *Xcc* strain 049, its TAL-free mutant *Xcc* 049E, and *Xcc* 049E that expressed *pthA4* (*Xcc*049E/*pthA4*) were inoculated on *C. sinensis* plants. The TAL-free mutant 049E caused extremely weak canker symptoms, which were restored by expressing *pthA4* in *Xcc* 049E (*Figure 5E*). No transcript of *pthA4* was detected in the plants inoculated with *Xcc* 049E. In the plants inoculated with *Xcc* 049 or *Xcc* 049E/*pthA4*, the levels of expression of *pthA4* and *CsLOB1* were enhanced; moreover, the transcript level at 10 dpi was higher than that at 5 dpi (*Figure 5E*). In contrast, the level of expression of *Cs9g12620* at 10 dpi was distinctively lower than that at 5 dpi (*Figure 5E*). By comparison with 5 dpi, the cell number of *Xcc* 29-1 in planta was increased

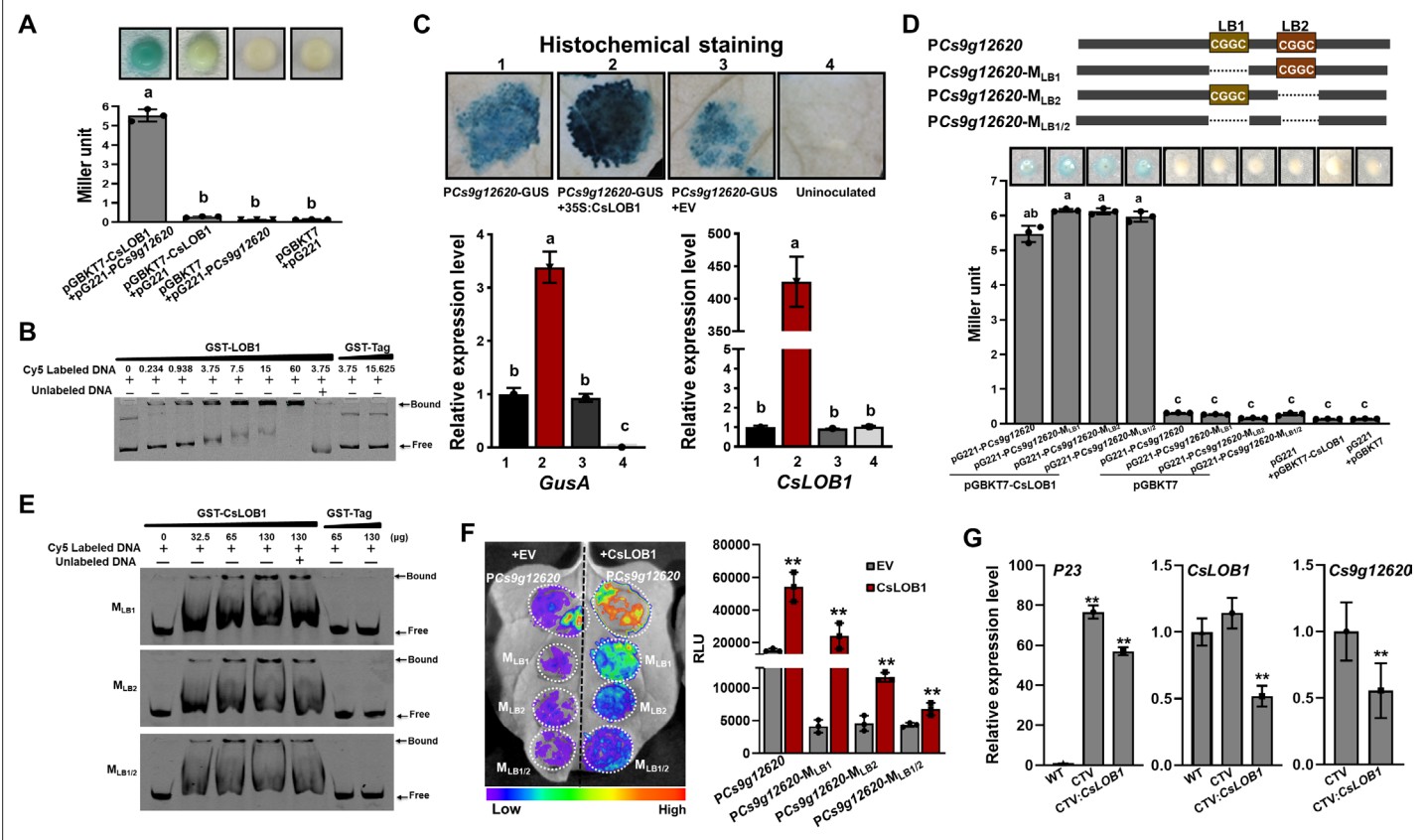

**Figure 4.** The *Cs9g12620* promoter is directly activated by CsLOB1. (**A**) A yeast one-hybrid (Y1H) assay shows that CsLOB1 interacts with the *Cs9g12620* promoter. pGBKT7-CsLOB1 and pG221-P*Cs9g12620* were co-transformed into yeast strain EGY48 and screened on synthetic dropout SD/-Ura and SD/-Ura/-Trp media. The transformed cells were dissociated by repeated freeze–thaw treatment and then used for β-galactosidase assays on filters soaked with Z buffer that contained 20 µg/ml of X-gal. pGBKT7 and pG221 vectors were used as the negative control by co-transformation with corresponding constructs. The columns below show the quantitative assay of β-galactosidase by the Miller method. Values are the means ± SD (*n* = 3 biological replicates). Columns labeled with different letters indicate significant difference among means (ANOVA, p < 0.01). (**B**) An electrophoretic mobility shift assay (EMSA) shows that CsLOB1 binds to the *Cs9g12620* promoter. Purified GST-CsLOB1 was incubated with 25 ng of promoter DNA in gel shift binding buffer at 28°C for 30 min and was then analyzed by 6% non-denaturing PAGE. (**C**) *Cs9g12620* promoter activity is induced by CsLOB1. CsLOB1 was transiently co-expressed with the *Cs9g12620* promoter GUS fusion in *Nicotiana benthamiana*. Promoter-driven GUS activity was assayed at 2 days post-agroinfiltration. The upper images show the histochemical staining of *Cs9g12620* promoter-driven GUS activity. The bar chart below shows the qRT-PCR analysis of transcript levels of *gusA* and *pthA4*. The level of expression of each gene in the samples that expressed the promoter alone was set as 1, and the levels of the other samples were calculated relative to that. Values are the mean results from three biological replicates and are the means ± SD. Columns labeled with different letters indicate significant difference among means (ANOVA, p < 0.01). (**D**) A Y1H assay shows that CsLOB1 interacts with the *Cs9g12620* promoter P*Cs9g12620* and CsLOB1-binding site deletion mutants P*Cs9g12620*-M$_{LB1}$, P*Cs9g12620*-M$_{LB2}$, and P*Cs9g12620*-M$_{LB1/2}$. The upper diagram shows the putative CsLOB1-binding sites and the corresponding deletion mutants. The bar chart below represents a Y1H assay that shows the interaction of CsLOB1 with the promoter mutants. The analyses were same as those conducted in **A**. Columns labeled with different letters indicate significant difference among means (ANOVA, p < 0.01). (**E**) An EMSA showed that CsLOB1 binds to *Cs9g12620*-M$_{LB1}$, P*Cs9g12620*-M$_{LB2}$, and P*Cs9g12620*-M$_{LB1/2}$. Purified GST-CsLOB1 was incubated with 25 ng of promoter DNA in gel shift binding buffer at 28°C for 30 min. The analyses were same as those in **B**. (**F**) Luciferase assays show the role of CsLOB1 on P*Cs9g12620*-M$_{LB1}$, P*Cs9g12620*-M$_{LB2}$, and P*Cs9g12620*-M$_{LB1/2}$. Promoter luciferase fusions were co-expressed with CsLOB1 in *N. benthamiana*. Co-expression with the empty binary vector pHB was used as the control. Luciferase activity was measured with a CCD imaging system at 2 days post-agroinfiltration. The image on the right shows the quantification of luciferase signal using a microplate luminescence reader. Values are the means ± SD (*n* = 3 biological replicates, **p < 0.01 [Student's *t*-test]). (**G**) qRT-PCR analysis of the transcript level of *Cs9g12620* in *CsLOB1*-silenced plants. *CsLOB1* was silenced in *Citrus sinensis* using the citrus tristeza virus (CTV)-based gene silencing vector CTV33. The effective infection by CTV was evaluated by the expression of *P23* gene harbored in CTV33. The silencing efficiency of CsLOB1 was verified by comparison with the levels of expression in the WT and empty CTV33-infected control plants. The reduced expression of *Cs9g12620* is shown by comparison with the level of expression in the empty CTV33-infected plants. The qRT-PCR was performed as described in **C**. ANOVA, analysis of variance; CCD, charge-coupled device; GUS, β-glucuronidase; PAGE, polyacrylamide gel electrophoresis; qRT-PCR, real-time quantitative reverse transcription PCR; SD, standard deviation; WT, wild type.

The online version of this article includes the following source data for figure 4:

*Figure 4 continued on next page*

*Figure 4 continued*

**Source data 1.** Excel file containing LacZ, qRT, and RLU raw data in ***Figure 4A, C, D, F, G***.

**Source data 2.** Original file for the yeast one-hybrid assay in ***Figure 4A*** (pGBKT7-CsLOB1+pG221-P*Cs9g12620*, pGBKT7-CsLOB1+pG221, pGBKT7+pG221-P*Cs9g12620*, and pGBKT7+pG221).

**Source data 3.** PDF containing ***Figure 4A*** and original scans of the yeast one-hybrid assay (pGBKT7-CsLOB1+pG221-P*Cs9g12620*, pGBKT7-CsLOB1+pG221, pGBKT7+pG221-P*Cs9g12620*, and pGBKT7+pG221) with highlighted bands.

**Source data 4.** Original file for the electrophoretic mobility shift assay in ***Figure 4B*** (GST-LOB1+*PCs9g12620*).

**Source data 5.** PDF containing ***Figure 4B*** and original scans of the electrophoretic mobility shift assay (GST-LOB1+*PCs9g12620*) with highlighted bands.

**Source data 6.** Original file for histochemical staining of the *Cs9g12620* promoter-driven GUS activity in ***Figure 4C*** (P*Cs9g12620*-GUS, P*Cs9g12620*-GUS+35S:CsLOB1, P*Cs9g12620*-GUS+EV, and uninoculatied).

**Source data 7.** PDF containing ***Figure 4C*** and original scans of the histochemical staining of the *Cs9g12620* promoter-driven GUS activity assay (P*Cs9g12620*-GUS, P*Cs9g12620*-GUS+35S:CsLOB1, P*Cs9g12620*-GUS+EV, and uninoculatied) with highlighted bands.

**Source data 8.** Original file for the yeast one-hybrid assay in ***Figure 4D*** (pGBKT7-CsLOB1+*PCs9g12620*, pGBKT7-CsLOB1+P*Cs9g12620*-M$_{LB1}$, pGBKT7-CsLOB1+P*Cs9g12620*-M$_{LB2}$, and pGBKT7-CsLOB1+P*Cs9g12620*-M$_{LB1/2}$).

**Source data 9.** Original file for the yeast one-hybrid assay in ***Figure 4D*** (pGBKT7+*PCs9g12620*, pGBKT7+*PCs9g12620*-M$_{LB1}$, pGBKT7+P*Cs9g12620*-M$_{LB2}$, pGBKT7+P*Cs9g12620*-M$_{LB1/2}$, pGBKT7-CsLOB1+pG221, and pGBKT7+pG221).

**Source data 10.** PDF containing ***Figure 4D*** and original scans of the yeast one-hybrid assay (pGBKT7-CsLOB1+P*Cs9g*12620, pGBKT7-CsLOB1+P*Cs9g12620*-M$_{LB1}$, pGBKT7-CsLOB1+P*Cs9g12620*-M$_{LB2}$, pGBKT7-CsLOB1+P*Cs9g12620*-M$_{LB1/2}$, pGBKT7+P*Cs9g*12620, pGBKT7+P*Cs9g12620*-M$_{LB1}$, pGBKT7+P*Cs9g12620*-M$_{LB2}$, pGBKT7+P*Cs9g12620*-M$_{LB1/2}$, pGBKT7-CsLOB1+pG221, and pGBKT7+pG221) with highlighted bands.

**Source data 11.** Original file for the electrophoretic mobility shift assay in ***Figure 4E*** (GST-LOB1+P*Cs9g12620*-M$_{LB1}$ and GST-LOB1+P*Cs9g12620*-M$_{LB2}$).

**Source data 12.** Original file for the electrophoretic mobility shift assay in ***Figure 4E*** (GST-LOB1+P*Cs9g12620*-M$_{LB1/2}$).

**Source data 13.** PDF containing ***Figure 4E*** and original scans of the electrophoretic mobility shift assay (GST-LOB1+P*Cs9g12620*-M$_{LB1}$, GST-LOB1+P*Cs9g12620*-M$_{LB2}$, and GST-LOB1+P*Cs9g12620*-M$_{LB1/2}$) with highlighted bands.

**Source data 14.** Original file for the luciferase assays in ***Figure 4F*** (P*Cs9g12620*, *PCs9g12620*-M$_{LB1}$, *PCs9g12620*-M$_{LB2}$, and P*Cs9g12620*-M$_{LB1/2}$ co-expression with CsLOB1 and EV, respectively).

**Source data 15.** PDF containing ***Figure 4F*** and original scans of the luciferase assays (P*Cs9g12620*, *PCs9g12620*-M$_{LB1}$, *PCs9g12620*-M$_{LB2}$, and P*Cs9g12620*-M$_{LB1/2}$ co-expression with CsLOB1 and EV, respectively) with highlighted bands.

---

1.5-fold at 10 dpi (***Figure 5F***). However, the expressions of *pectin esterase* and *expansin* genes associated with canker development were decreased at 10 dpi (***Figure 5G***). This supported the hypothesis that the overexpression of PthA4 exerts a feedback inhibition on the expression of *Cs9g12620*, which was consistent with the result that the transient overexpression of PthA4 suppressed the *Cs9g12620* promoter activation by CsLOB1. The dynamic regulation of *Cs9g12620* is closely related with canker development.

## PthA4 interacts with CsLOB1

PthA4 exerted a role in suppressing the *Cs9g12620* promoter independently of its binding to EBE, which led us to determine whether PthA4 could physically interact with CsLOB1. In yeast two-hybrid (Y2H) assays, a positive interaction was observed in yeast strain AH109 co-transformed with AD-PthA4 and BD-CsLOB1 (***Figure 6A***). In a GST pull-down assay, GST-CsLOB1 successfully pulled down MBP-PthA4. In the control lane, the GST tag did not pull down MBP-PthA4 (***Figure 6B***). The interaction between PthA4 and CsLOB1 in vivo was further verified by split luciferase assays on *N. benthamiana* (***Figure 6C***).

## *Cs9g12620* is involved in the formation of canker symptoms

The *Cs9g12620*, as well as *CsLOB1* and *pthA4*, were transiently overexpressed in *C. sinensis*. The expression of *pthA4* induced a canker-like phenotype. The overexpression of *Cs9g12620* resulted in yellowing and hypertrophy of the leaves of *C. sinensis* plants (***Figure 7A***). The overexpression of *CsLOB1* did not cause a distinctive phenotype in *C. sinensis* plants. Thus, the overexpression of *Cs9g12620* elicited a canker-like symptom in *C. sinensis* that resembled that induced by *pthA4*, although the symptoms were weaker. Transmission electron microscopy (TEM) was conducted to investigate the changes in *C. sinensis* leaf structure. In the control leaves infiltrated with empty vector,

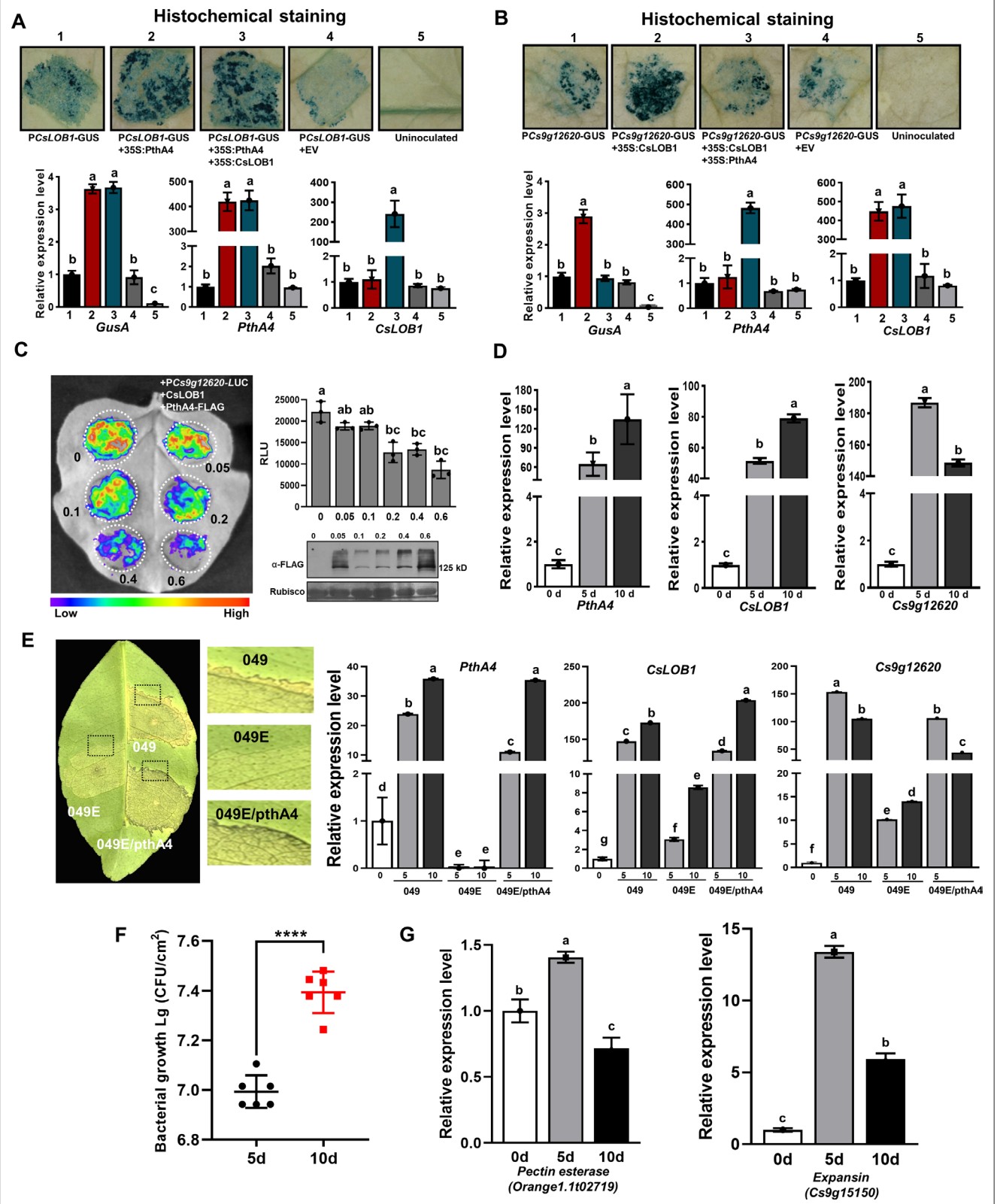

**Figure 5.** CsLOB1-induced *Cs9g12620* promoter activity is suppressed by PthA4. (**A**) PthA4 induced the activity of *CsLOB1* promoter. PthA4 was transiently co-expressed with the *CsLOB1* promoter GUS fusion in *Nicotiana benthamiana*. GUS activity was assayed at 2 days post-agroinfiltration. The upper images show the histochemical staining of *CsLOB1* promoter (P*CsLOB1*)-driven GUS activity. The bar chart below shows the qRT-PCR analysis of the transcript levels of *gusA*, *CsLOB1*, and *pthA4*. The transcript level of each gene in the samples that expressed P*CsLOB1*-GUS was set as

*Figure 5 continued on next page*

*Figure 5 continued*

1, and the levels of the other samples were calculated relative to that. Values are the mean results from three biological replicates and are the means ± SD. Columns labeled with different letters indicate significant difference among means (ANOVA, p < 0.01). (**B**) PhtA4 suppressed the CsLOB1-induced activity of the *Cs9g12620* gene promoter. The suppression was evaluated by the co-expression of PthA4 with CsLOB1 and *PCs9g12620*-GUS. Histochemical staining and a qRT-PCR analysis were performed as described in **A**. Columns labeled with different letters indicate significant difference among means (ANOVA, p < 0.01). (**C**) PthA4 suppressed the CsLOB1-induced P*Cs9g12620* activity in a dose-dependent manner. CsLOB1 and P*Cs9g12620*-LUC were co-expressed with PthA4-FALG in *N. benthamiana*. The agrobacteria that expressed PthA4-FLAG were arranged to a cell concentration series of $OD_{600}$ values of 0.05, 0.1, 0.2, 0.4, and 0.6. The luciferase signal was quantified with a microplate luminescence reader. Values are the means ± SD (*n* = 3 biological replicates). Columns labeled with different letters indicate significant difference among means (ANOVA, p < 0.01). The image on the bottom right shows the expression of PthA4-FLAG by immunoblotting with anti-FLAG. (**D**) A qRT-PCR analysis of the levels of expression of *pthA4*, *CsLOB1*, and *Cs9g12620* in *Citrus sinensis* leaves inoculated with *Xcc* 29-1. The cell suspension ($10^8$ CFU/ml) was infiltrated into the plant leaves. qRT-PCR was performed at 0, 5, and 10 days post-inoculation (dpi). The level of expression of each gene at 0 dpi was set as 1, and the level in other samples was calculated relative to those baseline values. Values are the mean results from three biological replicates and are the means ± SD. Columns labeled with different letters indicate significant difference among means (ANOVA, p < 0.01). (**E**) The level of expression of *Cs9g12620* was dynamically related with that of PthA4 during *Xcc* infection. The disease symptoms caused by WT *Xcc* 049, transcriptional activator-like (TAL)-free mutant 049E, and *Xcc* 049E/*pthA4* on *C. sinensis* leaves. The phenotype was recorded at 10 dpi. A qRT-PCR analysis was conducted to evaluate the levels of expression of *pthA4, CsLOB1*, and *Cs9g12620*. The experiment was performed as described in **D**. Columns labeled with different letters indicate significant difference among means (ANOVA, p < 0.01). (**F**) Growth of wildtype *Xcc* 29-1 in *C. sinensis* plants. The *Xcc* 29-1 cells of $10^8$ CFU/ml were infiltrated into citrus leaves. At 5 and 10 days post inoculation, bacteria were recovered from leaves and counted on Nutrient Agar (NA) plates. Error bars represent the standard deviation from three independent experiments (Student's *t*-test, ****p < 0.0001). (**G**) qRT-PCR analysis of *pectin esterase (orange1.1t02719)* and *expansin (cs9g15150)* genes expression levels in *C. sinensis* leaves inoculated with *Xcc* 29-1. The experiment was performed as described in **D**. Columns labeled with different letters indicate significant difference among means (ANOVA, p < 0.01). ANOVA, analysis of variance; GUS, β-glucuronidase; qRT-PCR, real-time quantitative reverse transcription PCR; SD, standard deviation.

The online version of this article includes the following source data for figure 5:

**Source data 1.** Excel file containing qRT, RLU, and bacterial growth raw data in *Figure 5A–G*.

**Source data 2.** Original file for histochemical staining of the *CsLOB1* promoter-driven GUS activity in *Figure 5A* (P*CsLOB1*-GUS, P*CsLOB1*-GUS+35S:PthA4, P*CsLOB1*-GUS+35S:PthA4+35S:CsLOB1, P*CsLOB1*-GUS+EV, and uninoculatied).

**Source data 3.** PDF containing *Figure 5A* and original scans of the histochemical staining of the *CsLOB1* promoter-driven GUS activity assay (P*CsLOB1*-GUS, P*CsLOB1*-GUS+35S:PthA4, P*CsLOB1*-GUS+35S:PthA4+35S:CsLOB1, P*CsLOB1*-GUS+EV, and uninoculatied) with highlighted bands.

**Source data 4.** Original file for histochemical staining of the *Cs9g12620* promoter-driven GUS activity in *Figure 5B* (P*Cs9g12620*-GUS, P*Cs9g12620*-GUS+35S:CsLOB1, P *Cs9g12620*-GUS+35S:CsLOB1+35S:PthA4, P*Cs9g12620*-GUS+EV, and uninoculatied).

**Source data 5.** PDF containing *Figure 5B* and original scans of the histochemical staining of the *Cs9g12620* promoter-driven GUS activity assay (P*Cs9g12620*-GUS, P*Cs9g12620*-GUS+35S:CsLOB1, P *Cs9g12620*-GUS+35S:CsLOB1+35S:PthA4, P*Cs9g12620*-GUS+EV, and uninoculatied) with highlighted bands.

**Source data 6.** Original file for the luciferase assays in *Figure 5C* (P*Cs9g12620*+CsLOB1+PthA4-FLAG).

**Source data 7.** PDF containing *Figure 5C* and original scans of the luciferase assays (P*Cs9g12620*+CsLOB1+PthA4-FLAG) with highlighted bands.

**Source data 8.** Original file for the western blot analysis in *Figure 5C* (anti-FLAG).

**Source data 9.** Original file for the western blot analysis in *Figure 5C* (Rubisco).

**Source data 10.** PDF containing *Figure 5C* and original scans of the western blot analysis (anti-FLAG and Rubisco) with highlighted bands.

**Source data 11.** Original image for the disease symptoms analysis in *Figure 5E* (049, 049E, and 049E/pthA4).

**Source data 12.** PDF containing *Figure 5E* and original scans of the disease symptoms analysis (049, 049E, and 049E/pthA4) with highlighted bands.

large air spaces were found between the cells in spongy mesophyll. In comparison, more cells and less air space were found in the spongy mesophyll of the leaves that transiently expressed *Cs9g12620* and *pthA4* (*Figure 7B*).

The contribution of *Cs9g12620* to canker development was subsequently examined by CTV-induced gene silencing in *C. sinensis* plants. The efficiency of gene silencing was evaluated in 1-month-old leaves at 60 days post-stem inoculation. The infection by CTV was indicated by verification of the transcripts of CTV *P23* gene. The transcript level of *Cs9g12620* was then verified to be reduced by approximately 70% in the gene-silenced *C. sinensis* plants (*Figure 7C*). Inoculation of the WT *Xcc* 29-1 caused remarkably attenuated canker symptoms in the *Cs9g12620*-silenced plants. In comparison with the control plants that were infected with empty CTV, there were remarkably fewer symptoms on the *Cs9g12620*-silenced plants (*Figure 7D*). The *Xcc* 29-1 in the *Cs9g12620*-silenced plants grew more slowly than in the control plants. Significantly fewer cell numbers were isolated from the *Cs9g12620*-silenced plants than from the control plants at 2, 4, and 6 days post-inoculation

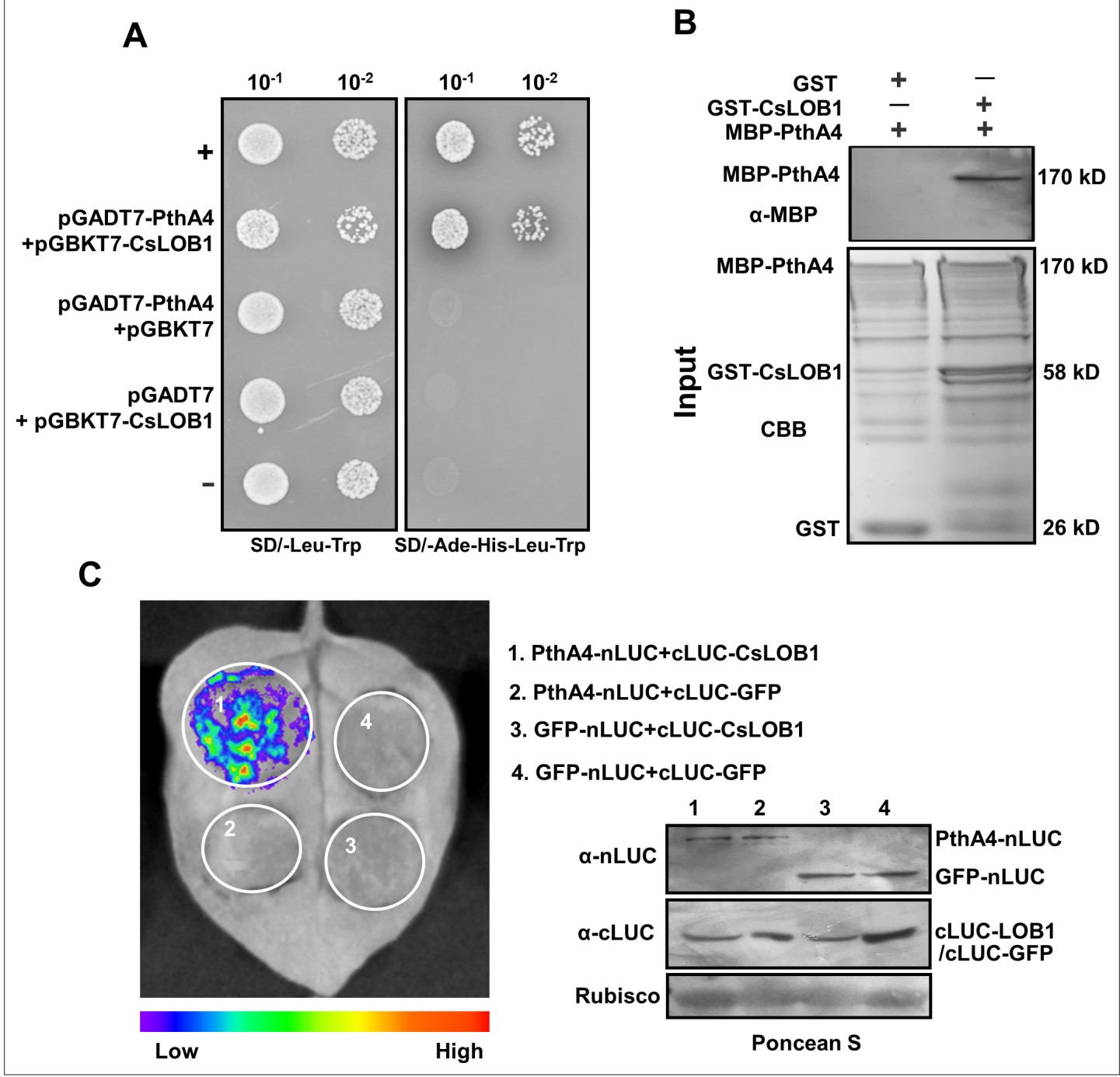

**Figure 6.** PhtA4 interacts with CsLOB1. (**A**) A yeast two-hybrid assay shows that PhtA4 interacts with CsLOB1. pGADT7-PthA4 and pGBKT7-CsLOB1 were co-transformed in yeast cells and screened on synthetic dextrose media that lacked leucine and tryptophan (SD/-Leu-Trp). A single yeast colony was then selected for serial dilution and grown on SD/-Leu-Trp and SD/-Ade-His-Leu-Trp to examine the interaction. The yeast co-transformed with pGADT7-T and pGBKT7-53 served as a positive control. The yeast co-transformed with pGADT7-T and pGBKT7-lam served as a negative control. (**B**) GST pull-down assay of the interaction between PhtA4 and CsLOB1. The recombinant GST-CsLOB1 and MBP-PthA4 proteins were used for GST pull-down assays. The GST protein served as a negative control. The pull-down of MBP-PhtA4 was verified by anti-MBP immunoblotting. The experiment was repeated three times with similar results. (**C**) A split luciferase assay for the interaction of PhtA4 and CsLOB1 in vivo. Nicotiana benthamiana leaves were co-infiltrated with agrobacteria that harbored 35S:PhtA4-nLUC and 35S:cLUC-CsLOB1. The image on the right shows the expression of respective proteins. Images of chemiluminescence were obtained by treatment with 0.5 µM luciferin 48 hr post-infiltration. Similar results were obtained in three biological repeats. GUS, β-glucuronidase.

The online version of this article includes the following source data for figure 6:

*Figure 6 continued on next page*

*Figure 6 continued*

**Source data 1.** Original file for the yeast two-hybrid assay in *Figure 6A* (PthA4+CsLOB1, SD/-Leu-Trp).

**Source data 2.** Original file for the yeast two-hybrid assay in *Figure 6A* (PthA4+CsLOB1, SD/-Ade-His-Leu-Trp).

**Source data 3.** PDF containing *Figure 6A* and original scans of the yeast two-hybrid assay (PthA4+CsLOB1, SD/-Leu-Trp and SD/-Ade-His-Leu-Trp) with highlighted bands.

**Source data 4.** Original file for the western blot analysis in *Figure 6B* (anti-MBP).

**Source data 5.** Original file for the Coomassie Brilliant Blue (CBB) analysis in *Figure 6B*.

**Source data 6.** PDF containing *Figure 6B* and original scans of the western blot analysis (anti-FLAG and CBB) with highlighted bands.

**Source data 7.** Original file for the luciferase assays in *Figure 6C* (PthA4-nLUC+cLUC-CsLOB1, PthA4-nLUC+cLUC-GFP, GFP-nLUC+cLUC-CsLOB1, and GFP-nLUC+cLUC-GFP).

**Source data 8.** PDF containing *Figure 6C* and original scans of the luciferase assays (PthA4-nLUC+cLUC-CsLOB1, PthA4-nLUC+cLUC-GFP, GFP-nLUC+cLUC-CsLOB1, and GFP-nLUC+cLUC-GFP) with highlighted bands.

**Source data 9.** Original file for the western blot analysis in *Figure 6C* (anti-nLUC and anti-cLUC).

**Source data 10.** Original file for the western blot analysis in *Figure 6C* (Rubisco).

**Source data 11.** PDF containing *Figure 6C* and original scans of the western blot analysis (anti-nLUC, anti-cLUC, and Rubisco) with highlighted bands.

(*Figure 7E*). The expression levels of *Cs9g12620* and *CsLOB1* genes in *Cs9g12620*-silenced plants inoculated with *Xcc* 29-1 showed that the induced transcription levels of both were decreased, and the expression levels of *pthA4* were also significantly decreased, downregulated by 38%, 15%, and 16%, respectively (*Figure 7F*). The TAL-free 049E caused extremely wreak canker symptom, and wild-type 049 and 049E/*pthA4* caused severe canker on control plants. On *Cs9g12620*-silenced plants, wildtype 049 and 049E/*pthA4* caused greatly reduced canker symptom, while the 049E/*pthA4* caused weak canker symptom alike to TAL-free mutant 049E (*Figure 7G*). Furthermore, the population of 049 and 049E/*pthA4* recovered from *Cs9g12620*-silenced plants was lower than that in control plants. In contrast, population of 049E was low in either *Cs9g12620*-silenced or control plants (*Figure 7H*). This demonstrated expression of *pthA4* in 049E did not restore bacterial virulence on *Cs9g12620*-silenced plants. The level of expression of *Cs9g12620* is correlated with the formation of canker symptoms in citrus plants.

## Discussion

The ectopic expression of *avrXa7* in *Xcc* 29-1 did not induce canker symptoms on citrus plants (*Sun et al., 2018*), which was consistent with a previous study that reported that the browning phenotype was elicited by an *Xcc* L2 strain that expressed *avrXa7* (*Ishihara et al., 2003*). This was hypothesized to be caused by two groups of differentially expressed genes, including (1) resistance genes induced by AvrXa7, and (2) susceptibility genes induced by PthA4 but suppressed by the expression of AvrXa7. A transcriptome analysis was utilized to identify the different relevant citrus genes and showed that AvrXa7 specifically activated the aspartic protease gene *Cs7g06780.1* and linoleate 13S-lipoxygenase gene *orange1.1t03773.1* (*Sun et al., 2018*). In this study, we focused on identification of the genes induced by PthA4 for canker formation. Based on the pattern of expression and analysis of promoter activity, Cs9g12620 was proven to be dynamically regulated by the PthA4-mediated induction of CsLOB1 for the development of canker symptoms. This supported the hypothesis that the PthA4-mediated induction of CsLOB1 serves a critical role in the complicated regulatory network on gene expression during *Xcc* infection.

The original purpose of this study was to explore the genes that were directly targeted by PthA4 through TAL effector–DNA recognition. Based on a sequence analysis, the two genes *Cs9g12620* and *CsLOB1* were predicted to contain PthA4 EBEs. An analysis of the pattern of expression revealed that *Cs9g12620* and *CsLOB1* were induced by *Xcc* infection in a PthA4-dependent manner. Because the susceptibility gene *CsLOB1* has already been proven to be targeted by *Xcc* PthA4, the binding and regulatory role of PthA4 on the *Cs9g12620* promoter was assayed experimentally. Y1H, EMSA, and promoter reporter fusion analyses were used and enabled the conclusion that PthA4 bound to the *Cs9g12620* promoter, whereas it suppressed the promoter activity. Since two CsLOB1 putative binding sites were predicted from the *Cs9g12620* promoter, we demonstrated that CsLOB1 bound directly

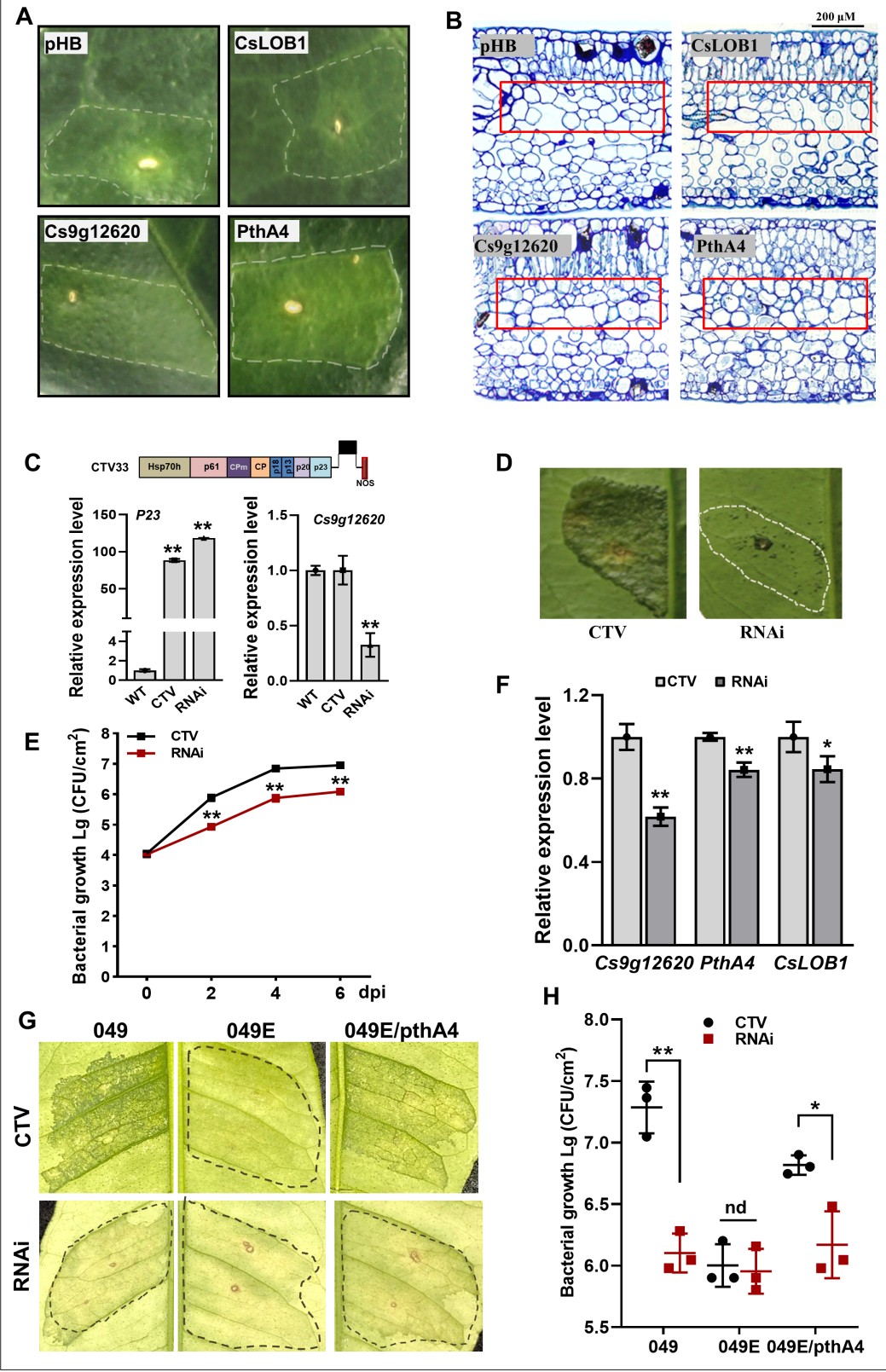

**Figure 7.** Cs9g12620 is involved in the formation of canker symptoms. (**A**) The transient overexpression of *Cs9g12620* results in hypertrophy in the *Citrus sinensis* leaves. The expression of *pthA4* was used as the positive control. The cultured *Agrobacterium* cells were adjusted to an OD$_{600}$ of 0.01 and infiltrated into young *C. sinensis* leaves. The phenotypes were recorded at 15 days post-agroinfiltration. The area of infiltration for each gene is

*Figure 7 continued on next page*

*Figure 7 continued*

indicated by a dotted line. (**B**) Transmission electron micrograph of *C. sinensis* leaf tissues. Transmission electron micrographs of the cross sections of citrus leaf tissue. The leaves were sampled at 15 days post-agroinfiltration. The changes in tissue structure induced by *Cs9g12620* and *pthA4* are indicated by a box. The bars represent 200 µM. (**C**) A real-time quantitative reverse transcription PCR (qRT-PCR) analysis of the silencing of *Cs9g12620* in *C. sinensis*. *Cs9g12620* was silenced using the citrus tristeza virus (CTV)-based gene silencing vector CTV33. The effective infection by CTV was evaluated by measuring the level of expression of the *P23* gene harbored in CTV33. The silencing efficiency of *Cs9g12620* was verified by comparison with the levels of expression in the WT and empty CTV33-infected control plants. Values are the mean results from three biological replicates (Student's *t*-test, **p < 0.01). The experiment was repeated three times. (**D**) The canker symptoms on *Cs9g12620*-silenced citrus plants were caused by the WT *Xcc* 29-1. The *Xcc* 29-1 cells of $10^7$ CFU/ml were infiltrated into citrus leaves to analyze the development of canker symptoms. The canker symptoms were recorded at 6 days post-inoculation (dpi). The areas of infiltration are indicated by dotted lines. The inoculation of the plants infected by the empty vector CTV33 was used as the negative control. (**E**) Growth of *Xcc* 29-1 in *Cs9g12620*-silenced citrus plants. A volume of $10^7$ CFU/ml of *Xcc* 29–1 cells were infiltrated into citrus leaves. At 2, 4, and 6 dpi, bacteria were recovered from leaves and counted on Nutrient Agar (NA) plates. Error bars represent the standard deviation from three independent experiments (Student's *t*-test, *p < 0.05, **p < 0.01). (**F**) A qRT-PCR analysis of *Cs9g12620*, *CsLOB1*, and *pthA4* in *Cs9g12620*-silenced citrus plants inoculated with *Xcc* 29-1. The experiment was performed as described in **C**. (**G**) The canker symptoms on *Cs9g12620*-silenced citrus plants were caused by the WT *Xcc* 049, transcriptional activator-like (TAL)-free mutant 049E, and 049E/PthA4. The experiment was performed as described in **D**. (**H**) Growth of WT *Xcc* 049, 049E, and 049E/PthA4 in *Cs9g12620*-silenced citrus plants. At 6 dpi, bacteria were recovered from leaves and counted on nutrient-rich (NA) plates. The experiment was performed as described in **E**. dpi, days post-inoculation; WT, wild type.

The online version of this article includes the following source data for figure 7:

**Source data 1.** Excel file containing qRT and bacterial growth raw data in *Figure 7C, E, F, H*.

**Source data 2.** Original image of transient overexpression (pHB, CsLOB1, Cs9g12620, and PthA4) in the *C. sinensis* leaves in *Figure 7A*.

**Source data 3.** PDF containing *Figure 7A* and original scans of the transient overexpression analysis (pHB, CsLOB1, Cs9g12620, and PthA4) with highlighted bands.

**Source data 4.** Original image of transmission electron micrograph of *C. sinensis* leaf tissues transient overexpression pHB in *Figure 7B*.

**Source data 5.** Original image of transmission electron micrograph of *C. sinensis* leaf tissues transient overexpression CsLOB1 in *Figure 7B*.

**Source data 6.** Original image of transmission electron micrograph of *C. sinensis* leaf tissues transient overexpression Cs9g12620 in *Figure 7B*.

**Source data 7.** Original image of transmission electron micrograph of *C. sinensis* leaf tissues transient overexpression PthA4 in *Figure 7B*.

**Source data 8.** Original image for the disease symptoms analysis in *Figure 7D* (citrus tristeza virus, CTV).

**Source data 9.** Original image for the disease symptoms analysis in *Figure 7D* (RNAi).

**Source data 10.** Original image for the disease symptoms analysis in *Figure 7G* (citrus tristeza virus, CTV).

**Source data 11.** Original image for the disease symptoms analysis in *Figure 7G* (RNAi).

**Source data 12.** PDF containing *Figure 7G* and original scans of the disease symptoms analysis (citrus tristeza virus [CTV] and RNAi) with highlighted bands.

to the *Cs9g12620* promoter. Most importantly, CsLOB1 could activate the *Cs9g12620* promoter. As such, *Cs9g12620* is actually a target of *CsLOB1*.

The ectopic expression of *CsLOB1* restores the mutant canker symptoms of *Xcc pthA4*, which demonstrates that *CsLOB1* is the key target of PthA4 and thus, induces pustule symptoms (*Duan et al., 2018*). The transient overexpression of CsLOB1 did not induce an obvious phenotype in the citrus plant leaves, which was consistent with the findings of previous research (*Hu et al., 2014*). Instead, the contribution of CsLOB1 to canker symptoms has been verified by the transformation of 'Duncan' grapefruit (*Citrus* x *paradisi*) plants with a 35S:CsLOB1-glucocorticoid receptor construct, which suggested that the continuous expression of CsLOB1 is essential for the development of canker symptoms (*Duan et al., 2018*). *Duan et al., 2018* reported that a number of citrus genes appear to be associated with *CsLOB1*, but only the *Cs2g20600* promoter probe interacted with CsLOB1.

*Cs2g20600* encodes a zinc finger C3HC4-type RING finger protein with E3 ubiquitin ligase activity (***Duan et al., 2018***). In this study, we found that *Cs9g12620* was another target that is directly regulated by CsLOB1. The regulatory role of *CsLOB1* includes promoting host cell proliferation, regulating citrus cell wall remodeling, and affecting hormone signaling pathways, which is accompanied by the optimal growth of *Xcc* (***Zou et al., 2021***). Although it has been reported to specifically bind to -CGGC- in its target promoter region, CsLOB1 retained the ability to bind the *Cs9g12620* promoter DNA when the two specific binding sites LB1 and LB2 were deleted. We hypothesized that there are other putative binding sites of CsLOB1 in the *Cs9g12620* promoter, and multiple citrus genes are directly targeted by CsLOB1 for the development of cankers during *Xcc* infection.

PthA4 acts as a sole disease TAL required by *Xcc* to cause symptoms of canker on citrus plants. It localizes to the nuclei following secretion into host cell by the T3SS. A transient overexpression analysis showed that PthA4 exhibited a particular suppressive effect on the CsLOB1-activated *Cs9g12620* promoter activity. Since the PthA4-binding site is critical for activity of the *Cs9g12620* promoter, the binding of PthA4 suppressed the promoter activity. This prompted us to study whether PthA4 affects CsLOB1 alternatively. Y2H and pull-down assays showed that PthA4 interacted with CsLOB1 at the protein–protein level. This provided a new insight to study the dynamical regulatory role of PthA4 during *Xcc* infection since we found that PthA4 affects the expression of *Cs9g12620* in a dose-dependent manner. When PthA4 was expressed at a high level at 10 dpi, the level of expression of *Cs9g12620* was surprisingly lower. This suggested that there is a precise mechanism of the PthA4-CsLOB1-*Cs9g12620* regulatory cascade during infection with *Xcc*.

Although the amino acid sequence of Cs9g12650 was 86.3% homologous with that of Cs9g12620, they differed in their promoter sequences. The promoter of *Cs9g12650* lacks 22 bp of nucleotides at its 3' terminus, which are critical for the activity of its promoter. The activity of *Cs9g12620* promoter was completely lost if the 22 bp sequence was deleted by mutation. Therefore, *Cs9g12650* is not transcribed in citrus plants. This supported the hypothesis that the null activity of *Cs9g12650* promoter was owing to genetic diversity in citrus plants, which leads to different patterns of expression between the two paralogs in citrus. The 22 bp sequence at the *Cs9g12620* promoter could be a potential site for gene editing to create citrus plants that are resistant to citrus canker. A similar regulation of expression has been found in *CsLOB1*; CsLOB2 and CsLOB3 show the same ability to induce pustule formation as CsLOB1, but their promoters do not contain the PthA4 EBE; therefore, neither can be induced by PthA4 (***Zhang et al., 2017***).

The Cs9g12620 protein contains three distinctive conserved domains, including a signal peptide, a duplicated DUF642 region, and a bacterial exosortase protein domain. The duplicated DUF642 region has been shown to be a carbohydrate-binding domain. The exosortase system is involved in directing transport across the plasma membrane, with additional posttranslational modifications, such as glycosylation (***Haft et al., 2012***). This system is distributed across many biofilm-producing environmental bacteria and is closely related to the *N*-acyl amino acid synthases in Proteobacteria (***Craig et al., 2011***). It was hypothesized that Cs9g12620 could possibly be involved in the transportation of carbohydrates in citrus cells. This was similar to the SWEET genes induced by the effectors that pathogenic bacteria secrete into plants, and they export intracellular sugar into the extracellular space, which can be utilized by pathogenic bacteria (***Hu et al., 2014***). The binding of carbohydrates and the exact role for canker formation merits further study.

In conclusion, the putative carbohydrate-binding protein gene *Cs9g12620* was identified as a target of CsLOB1, which was directly regulated by PthA4-mediated induction of CsLOB1. In addition to the enhanced pattern of expression in response to *Xcc* infection, *Cs9g12620* was dynamically feedback regulated by PthA4 in a dose-dependent manner. At the early infection stage, the presence of PthA4 is responsible for the induction of CsLOB1, which is necessary for the activation of Cs9g12620. At the late infection stage, the over-redundancy of PthA4 exerts a feedback suppression effect on the activation of *Cs9g12620* by CsLOB1 (***Figure 8***). The findings provide strong evidence to understand the molecular complexity of PthA4-mediated induction of CsLOB1 during the development of citrus canker.

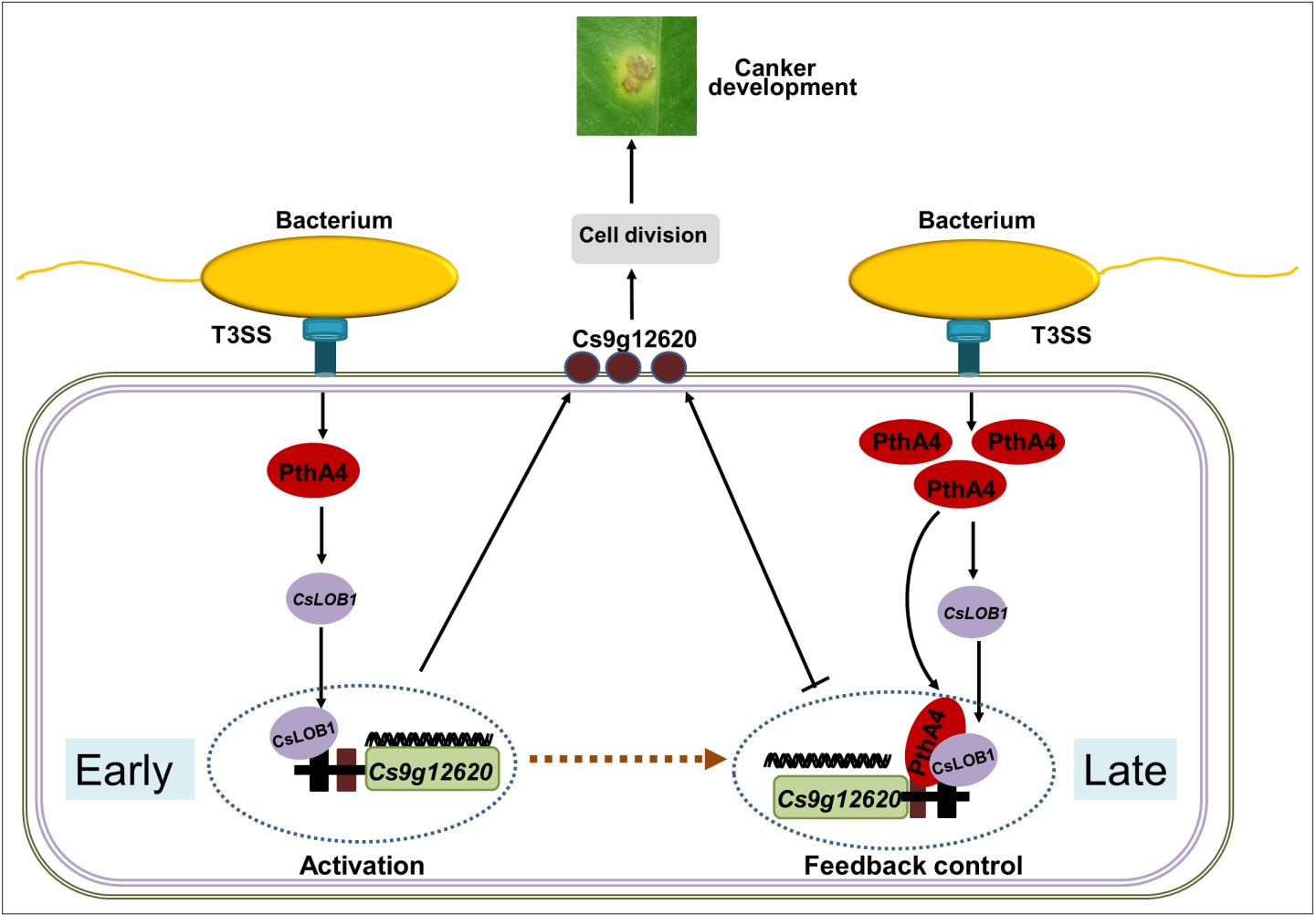

**Figure 8.** Proposed model of the expression of *Cs9g12620* directed by PthA4-mediated induction of CsLOB1. The regulation of putative carbohydrate-binding protein gene *Cs9g12620* was divided into two steps. At the early infection stage, the presence of PthA4 is responsible for the induction of CsLOB1, which is responsible for the induction of *Cs9g12620*. At the late infection stage, the over-redundancy of PthA4 exerts a feedback suppression effect on the activation of *Cs9g12620* by interacting with CsLOB1.

## Materials and methods
### Plant and bacterial materials

*C. sinensis* and *N. benthamiana* were grown in a greenhouse (26/24°C light/dark) under long-day conditions (16/8 hr light/dark). The strains and plasmids used in this study are listed in *Supplementary file 1a*. *Xcc* strain 29-1 was collected from *C. sinensis* in Jiangxi Province, China (*Ye et al., 2013*). The *Xcc* 29-1/avrXa7 strain is a *Xcc* 29-1 strain that expresses *avrXa7* from *Xanthomonas oryzae* pv. *oryzae*, the causal agent of rice bacterial blight (*Sun et al., 2018*). Mxac126-80 is a *pthA4* mutant with a Tn5 insertion at the 1789 bp position in the middle of RVD coding sequence (*Song et al., 2015*). *Xcc* strain 049 was collected from *C. sinensis* in Chongqing Province, China (*Ye et al., 2013*). *Xcc* 049E is a TAL-free strain of *Xcc* 049, and *Xcc* 049E/pthA4 is *Xcc* 049E that expresses the *pthA4* gene (*Ge et al., 2019*). All the *Xcc* strains were cultured in nutrient broth (NB) or nutrient agar (NA, 1.5% agar) at 28°C (*Song et al., 2015*). *Agrobacterium tumefaciens* strain GV3101 was cultured in YEP medium (10 g/l yeast extract, 10 g/l peptone, and 5 g/l sodium chloride, pH 7.0) at 28°C. The yeast strains EGY48 for the Y1H assays and AH109 for the Y2H assays were cultured at 30°C in YPD media (10 g/l yeast extract, 20 g/l peptone, and 20 g/l glucose, pH 4.5–5.0).

## Inoculation of *Xcc* in citrus host plants

The cultured *Xcc* cells were suspended in sterile distilled water to a final concentration of $10^8$ CFU/mL ($OD_{600}$ = 0.3). The bacterial suspension was inoculated onto fully expanded *C. sinensis* leaves with a needleless syringe (*Wu et al., 2019*). The canker disease symptoms were recorded at 5 or 10 dpi. To assess the bacterial growth in planta, 0.25 cm² leaf discs from the areas of infiltration were collected to count the cell numbers. The cell numbers were calculated as CFU per cm². All the experiments were repeated three times.

## Real-time qRT-PCR

qRT-PCR was performed using a CFX Connect Real Time PCR detection system (Bio-Rad, Shanghai, China) using iTaq Universal SYBR Green Supermix (Bio-Rad). The total RNA was isolated from *C. sinensis* and *N. benthamiana* leaves using a Plant RNA Kit (Omega, Shanghai, China). A volume of 2 µg of total RNA was reverse transcribed into single-stranded cDNA using AMV reverse transcriptase (TaKaRa, Dalian, China). The primer sequences used for qRT-PCR are listed in *Supplementary file 1b*. The *C. sinensis CsActin* and *N. benthamiana NbEF1α* genes were used as internal controls to statistically analyze the levels of relative expression (*Wu et al., 2019*). All the experiments were repeated three times.

## Luciferase assays

To study the interaction in vivo, the *PthA4* gene from *Xcc* was fused with nLUC at the N terminus. The *CsLOB1* gene from *C. sinensis* was fused with cLUC at the C terminus. cLUC-CsLOB1 was co-expressed with nLUC-PthA4 in 4-week-old *N. benthamiana* leaves. To analyze the promoter activity, the 463 bp *Cs9g12620* promoter region was directly introduced into vector pGWB435 by a Gateway cloning strategy to generate P*Cs9g12620*-LUC. Site-directed mutation of the first nucleotide 'T' in EBE of PthA4 or deletion of CsLOB1-binding sites LB1 and LB2 were created by overlapping PCR using the primers listed in *Supplementary file 1b*. The promoter mutants $M_A$, $M_C$, $M_G$, $M_{LB1}$, $M_{LB2}$, and $M_{LB1/2}$ that were generated were fused with luciferase (LUC) in pGWB435. The activities of LUC were assayed after transient overexpression in *N. benthamiana*. For each assay at 2 dpi, the inoculated leaves were treated with 0.5 mM luciferin, kept in the dark for 1 min to quench the fluorescence, and then used to capture luciferase luminescence images by a cooled charge-coupled device imaging apparatus (Roper Scientific, Trenton, NJ, USA). Each experiment had three replicates.

## Yeast one-hybrid assays

The pGBKT7-pthA4 and pGBKT7-CsLOB1 constructs were created by inserting the *pthA4* and *CsLOB1* gene into the pGBKT7 vector, respectively (*Supplementary file 1b*). A 463-bp DNA fragment of the *Cs9g12620* gene promoter was cloned into the pG221 vector, which resulted in pG221-P*Cs9g12620* (*Supplementary file 1a and b*). To examine the role of PthA4 and CsLOB1-binding sites in interaction, mutants $M_A$, $M_C$, $M_G$, $M_{LB1}$, $M_{LB2}$, and $M_{LB1/2}$ were cloned in pG221 (*Supplementary file 1b*). Mutants $M_A$, $M_C$, and $M_G$ were co-transformed with pGBKT7-pthA4, and mutants $M_{LB1}$, $M_{LB2}$, and $M_{LB1/2}$ were co-transformed with pGBKT7-CsLOB1 into yeast EGY48 using the standard LiAc-mediated yeast transformation (*Ye et al., 2004*). pG221 that harbored a *CYC1* core promoter was used for a transactivation assay in PthA4 and CsLOB1. The yeast transformants were screened on synthetic dropout SD/-Ura and SD/-Ura/-Trp media plates (*Ye et al., 2004*). The transformed cells were subjected to five freeze–thaw cycles using liquid nitrogen, spotted onto a sterile filter, and soaked with Z buffer that contained 20 µg/ml of X-gal. β-Galactosidase activity was scored within 8 hr of incubation. The β-galactosidase activity was quantified using the standard Miller method (*Marburg, 2016*). Each experiment was performed three times.

## Electrophoretic mobility shift assays

The binding of both PthA4 and CsLOB1 to the *Cs9g12620* promoter DNA (including $M_{LB1}$, $M_{LB2}$, and $M_{LB1/2}$) was detected by an Cy5-labeled probe EMSA as described previously (*Fan et al., 2020*). The binding of PthA4 to the *Cs9g12620* promoter DNA ($M_A$, $M_C$, and $M_G$) was visualized by ethidium bromide staining as part of EMSA. The GST-pthA4 and GST-CsLOB1 constructs were created by separately inserting the *pthA4* and *CsLOB1* genes into the pET41a(+) vector (*Supplementary file 1a*). GST-PthA4, GST-CsLOB1, and GST tag were expressed separately in *E. coli* BL21 (DE3) by induction

with 1.0 mM isopropyl-β-D-thiogalactopyranoside (IPTG). The purified GST-PthA4, GST-CsLOB1, and GST tag were subjected to ethidium bromide staining with concentration gradients by a twofold dilution. The proteins were incubated with 25 ng of promoter DNA in gel shift binding buffer (50 ng/μl Poly (dI•dC), 0.05 M Tris–HCl, 1.0 mM EDTA(Ethylene Diamine Tetraacetic Acid), 0.15 M KCl, and 1.0 mM dithiothreitol). The GST-PthA4 protein, GST-CsLOB1 protein, and GST tag were diluted into series that consisted of 0.117, 0.469, 1.875, 7.5, 30, and 120 μg; 0.234, 0.938, 3.75, 7.5, 15, and 60 μg; and 1.953, 7.813, 3.75, and 15.625 μg; respectively. After incubation at 28°C for 30 min, the mixtures were subjected to a DNA mobility analysis by electrophoresis in a 6% non-denaturing acrylamide gel. The DNA bands were visualized by fluorescence imaging using a Typhoon Trio Variable Mode Imager (GE Healthcare, Chicago, IL, USA). Analyses of the binding to promoter mutants $M_A$, $M_C$, and $M_G$ were conducted with the same strategy but using different GST-PthA4 protein series. Each experiment had three replicates.

### GUS activity assays

The *Cs9g12620* promoter region was cloned into the binary vector pCAMBIA1381 to fuse with the *gusA* gene (*Supplementary file 1b*). The coding sequences of *pthA4* and *CsLOB1* were individually cloned into the binary vector pHB to obtain pHB-pthA4 and pHB-CsLOB1, respectively. The promoter–GUS fusion was co-transformed with pHB-pthA4 or pHB-CsLOB1 to examine the promoter activity in *N. benthamiana*. β-Glucuronidase (GUS) activity was assayed by histochemical staining of the activity after 2 dpi (*Hu et al., 2014*). Moreover, the levels of expression of *gusA* in the leaf samples were evaluated by qRT-PCR to quantify the promoter activity (*Sun et al., 2020*). Each experiment had three replicates.

### Y2H assays

The coding sequences of *pthA4* and *CsLOB1* were cloned individually into both the pGADT7 and pGBKT7 vectors. AD-PthA4/BD-CsLOB1 was then transformed into the yeast strain AH109 and screened on synthetic drop-out media that lacked leucine and tryptophan (SD/-Leu-Trp). Single colonies were cultured, serially diluted, and grown on SD/-Leu-Trp and SD/-Ade-Leu-Trp-His media to examine the possible interaction. Additionally, pGADT7-T and pGBKT7-53 were used as positive controls, while pGADT7-T and pGBKT7-lam served as negative controls. Each experiment had three replicates.

### GST pull-down assay

*PthA4* was cloned into pMAL-c4x to express the MBP-PthA4 fusion protein. *CsLOB1* was cloned into pET-41a(+) to express the GST-CsLOB1 fusion protein. MBP-PthA4 and GST-CsLOB1 were expressed separately in *E. coli* BL21 (DE3) by induction with 1.0 mM IPTG. After the proteins were purified by Glutathione Sepharose 4 Fast Flow (GE Healthcare) and Amylose Resin (New England Biolabs, Ipswich, MA, USA) affinity chromatography, the GST pull-down assays were performed. Each experiment had three replicates.

### Western blotting analysis

Western blotting was performed to detect the levels of expression of proteins in *N. benthamiana*. Leaf samples were harvested at 2 dpi. The leaves were ground in liquid nitrogen and extracted in extraction/washing buffer (Roche, Basel, Switzerland). The proteins that were extracted were eluted with 3×Laemmli loading buffer, resolved by 10% sodium dodecyl sulfate–polyacrylamide gel electrophoresis, and subjected to western blotting using anti-FLAG (Abmart, Shanghai, China), anti-nLUC (Abmart), or anti-cLUC polyclonal antibody (Abmart). Each experiment had three replicates.

### Transient overexpression of the *C. sinensis* genes

The coding sequences of *Cs9g12620* were cloned into the binary vector pHB to obtain pHB-Cs9g12620. *A. tumefaciens* GV3101 cells were transformed with the resulting recombinant plasmids pHB-Cs9g12620, pHB-pthA4, and pHB-CsLOB1. For transient expression in *C. sinensis*, *A. tumefaciens* cells that harbored different constructs were cultured and grown overnight in YEP media with shaking. The cells were pelleted by centrifugation at 6000 rpm for 10 min at 4°C, suspended in infiltration medium (10 mM $MgCl_2$, 10 mM MES(2-Morpholinoethanesulfonic Acid), and 200 mM acetosyringone,

pH 5.7) to an OD$_{600}$ of 0.01 and gently injected into the young leaves. The phenotypes were scored at 15 dpi. Each experiment had three replicates.

## Transmission electron microscopy

To examine any changes in the leaf tissue, the agroinfiltrated *C. sinensis* leaves were sampled 15 dpi for observation by TEM. The samples were fixed using 3% glutaraldehyde in 0.1 M potassium phosphate buffer (pH 7.2) and incubated at room temperature for 4 hr. The tissues were washed twice with the same buffer and post-fixed in 2% osmium tetroxide in 0.1 M potassium phosphate buffer at room temperature for 4 hr (*Etxeberria et al., 2009*). The samples were then dehydrated in an ethanol series and embedded in Spurr's resin. Thin sections (80–100 nm thick) were cut with a diamond knife, collected on 200 mesh copper grids, stained with 2% uranyl acetate, and post-stained with lead citrate. Those sections were then examined under an electron microscope (SU8020; Hitachi, Ltd, Tokyo, Japan) in scanning transmission electron microscope mode. Each experiment had three replicates.

## Silencing of *Cs9g12620* in the citrus plants

CTV33 is a stable expression vector based on the CTV (*El-Mohtar and Dawson, 2014*). A 488-bp DNA fragment of *Cs9g12620* and a 264-bp DNA fragment of *CsLOB1* were cloned separately into the CTV33 vector to generate the gene silencing vector. The CTV virion was prepared in *N. benthamiana* and inoculated in *C. sinensis* using the bark-flap method as previously described (*El-Mohtar and Dawson, 2014*; *Hajeri et al., 2014*). Three citrus plants were inoculated for each CTV virion. The plants inoculated with the CTV33 empty vector were used as controls. The infection of CTV was evaluated by the level of expression of the *P23* gene harbored in the CTV33 vector. The RNA silencing efficiency of *Cs9g12620* and *CsLOB1* was evaluated by quantification of the transcript level in the leaves from newly emerging branches. The development of cankers was then assessed by infiltration with *Xcc* 29-1 inoculum of 10$^7$ CFU/ml.

## Bioinformatics analysis

Sequence similarity searches of the reference *C. sinensis* genome database (https://www.citrusgenomedb.org/) were performed using BLAST. Putative promoter sequences were predicted using the online promoter analysis program Neural Network Promoter Prediction (http://www.fruitfly.org/seq_tools/promoter.html). Putative PthA4 EBEs were predicted using DNAMAN 8.0 (*Sun et al., 2018*). Putative signal peptides were predicted in PrediSi (http://www.predisi.de/) (*Hiller et al., 2004*). The conserved domains were identified using the NCBI Conserved Domain Database.

## Acknowledgements

We thank Gongyou Cheng and Lifang Zou in Shanghai Jiaotong University (Shanghai, China) for their gifts of *Xcc* strain 049, TAL-free mutant 049E, and 049E/pthA4. We also thank Dingzhong Tang and Haitao Cui for advice on using the cooled CCD imaging apparatus at the Plant Immune Center in Fujian Agriculture and Forestry University (Fuzhou, China). We would like to thank MogoEdit (https://www.mogoedit.com) for its English editing during the preparation of this manuscript.

## Additional information

### Funding

| Funder | Grant reference number | Author |
| --- | --- | --- |
| National Natural Science Foundation of China | 31872919 | Huasong Zou |
| National Natural Science Foundation of China | 31801696 | Xiaojing Fan |
| National Natural Science Foundation of China | 31701752 | Tao Zhuo |

| Funder | Grant reference number | Author |
|--------|------------------------|--------|

The funders had no role in study design, data collection, and interpretation, or the decision to submit the work for publication.

## Author contributions

Xinyu Chen, Conceptualization, Data curation, Software, Formal analysis, Validation, Investigation, Methodology, Writing - original draft; Huasong Zou, Resources, Software, Supervision, Funding acquisition, Writing - review and editing; Tao Zhuo, Resources, Project administration; Wei Rou, Validation; Wei Wu, Formal analysis; Xiaojing Fan, Project administration

## Author ORCIDs

Xinyu Chen ⓘD http://orcid.org/0009-0004-2721-8111
Huasong Zou ⓘD https://orcid.org/0000-0001-5758-1975

Reviewer #1 (Public Review): https://doi.org/10.7554/eLife.91684.3.sa1
Author response https://doi.org/10.7554/eLife.91684.3.sa2

---

# Additional files

## Supplementary files

• Supplementary file 1. Supplementary file with additional data. (**a**) List of bacterial strains and plasmids used in this study. (**b**) List of primers used in this study.

• MDAR checklist

## Data availability

All data generated or analyzed during this study are included in the manuscript and supporting files.

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
