## [Editor Report · eLife assessment]

This **valuable** study provides new insight into potential subtle dynamics in effector biology. The data presented generally support the claims, but in some cases, significant controls are missing and so the overall work is currently **incomplete**. If the limitations can be addressed, this work should be of broad relevance for biologists interested in molecular plant-microbe interactions.

---

## [Referee Report · Reviewer #1 (Public Review)]

Previous Review:

The authors have identified the predicted EBE of PthA4 in the promoter of Cs9g12620, which is induced by Xcc. The authors identified a homolog of Cs9g12620, which has variations in the promoter region. The authors show PthA4 suppresses Cs9g12620 promoter activity independent of the binding action. The authors also show that CsLOB1 binds to the promoter of Cs9g12620. Interestingly, the authors show that PthA4 interacts with CsLOB1 at protein level. Finally, it shows that Cs9g12620 is important for canker symptoms. Overall, this study has reported some interesting discoveries and the writing is generally well done. However, the discoveries are affected by the reliability of the data and some flaws of the experimental designs.

Here are some major issues:

The authors have demonstrated that Cs9g12620 contains the EBE of PthA4 in the promoter region, to show that PthA4 controls Cs9g12620, the authors need to compare to the wild type Xcc and pthA4 mutant for Cs9g12620 expression. The data in Figure 1 is not enough.

The authors confirmed the interaction between PthA4 and the EBE in the promoter of Cs9g12620 using DNA electrophoretic mobility shift assay (EMSA). However, Fig. 2B is not convincing. The lane without GST-PthA4 also clearly showed mobility shift. For EMSA assay, the authors need also to include non-labeled probe as competitor to verify the specificity. The description of the EMSA in this paper suggests that it was not done properly. It is suggested the authors to redo this EMSA assay following the protocol: Electrophoretic mobility shift assay (EMSA) for detecting protein-nucleic acid interactions PMID: 17703195.

The authors also claimed that PthA4 suppresses the promote activity of Cs9g12620. The data is not convincing and also contradicts with their own data that overexpression of Cs9g12620 causes canker and silencing of it reduces canker considering PthA4 is required for canker development. The authors conducted the assays using transient expression of PthA4. It should be done with Xcc wild type, pthA4 mutant and negative control to inoculate citrus plants to check the expression of Cs9g12620.

Fig. 6 AB is not convincing. There are no apparent differences. The variations shown in B is common in different wild type samples. It is suggested that the authors to conduct transgenic instead of transient overexpression.

Gene silencing data needs more appropriate controls. Fig. D. seems to suggest canker symptoms actually happen for the RNAi treated. The authors need to make sure same amount of Xcc was used for both CTV empty vector and the RNAi. It is suggested a blink test is needed here.

Comments on revised version:

Point 1: Addressed well.

Point 2: The EMSA was reconducted with adding unlabeled DNA, however, the results are still not convincing. Firstly, in fig.3D lane 5, with the absence of unlabeled DNA, the shifted bound signal wasn't reduced significantly. Secondly, still in fig.3D lane 5, the free labeled DNA probe at the bottom of the gel didn't increase. Which together mean that the unlabeled DNA was unable to compete with the labeled DNA and the "bound" shifted bands might not be true positive.

Point 3: The authors didn't address the question clearly regarding the connection between the inhibition of Cs9g12620 promoter by PthA4 and the positive function of Cs9g12620 on citrus canker.

Point 4: The comment was not addressed. Fig.7A and B are not convincing. Firstly, no evidence shows the expression of transiently expressed genes. Secondly, hard to tell the difference in 7A. Thirdly, since CsLOB1 positively regulates Cs9g12620, why expressing of CsLOB1 is unable to cause phenotype, while expression of PthA4 does?

Point 5: addressed.

---

## [Author Response]

The following is the authors’ response to the original reviews.

**Reviewer #1:**
Point 1: The authors have demonstrated that Cs9g12620 contains the EBE of PthA4 in the promoter region, to show that PthA4 controls Cs9g12620, the authors need to compare to the wild type Xcc and pthA4 mutant for Cs9g12620 expression. The data in Figure 1 is not enough.

The data in Figure 1 D and E show a pthA4 Tn5 insertion mutant Mxac126-80 and the expression level of Cs9g12620 in citrus inoculated with the pthA4 mutant.

Point 2: The authors confirmed the interaction between PthA4 and the EBE in the promoter of Cs9g12620 using DNA electrophoretic mobility shift assay (EMSA). However, Figure 2B is not convincing. The lane without GST-PthA4 also clearly showed a mobility shift. For the EMSA assay, the authors need also to include a non-labeled probe as a competitor to verify the specificity. The description of the EMSA in this paper suggests that it was not done properly. It is suggested the authors redo this EMSA assay following the protocol: Electrophoretic mobility shift assay (EMSA) for detecting protein-nucleic acid interactions PMID: 17703195.

Thank you very much for your comments. We have re-conducted the EMSA analysis based on your suggestion. The DNA probe was labeled with Cy5, included a non-labeled probe as a competitor. (Figure 3 B and D; Figure 4B and E)

Point 3: The authors also claimed that PthA4 suppresses the promote activity of Cs9g12620. The data is not convincing and also contradicts with their own data that overexpression of Cs9g12620 causes canker and silencing of it reduces canker considering PthA4 is required for canker development. The authors conducted the assays using transient expression of PthA4. It should be done with Xcc wild type, pthA4 mutant, and negative control to inoculate citrus plants to check the expression of Cs9g12620.

We have detected Cs9g12620 expression in silencing citrus plants inoculated wild type Xcc 29-1. (Figure 7F)

Point 4: Figure 6 AB is not convincing. There are no apparent differences. The variations shown in B are common in different wild-type samples. It is suggested that the authors conduct transgenic instead of transient overexpression.

It has been proven that transient expression of PthA4 leads to canker-like phenotype, suggesting that this experiment is effective. However, it will be more confident if conduct transgenic plant overexpressing pthA4 and Cs9g12620. We’ll create the plants in our following research to confirm the phenotype.

Point 5: Gene silencing data needs more appropriate controls. Figure D seems to suggest canker symptoms actually happen for the RNAi treated. The authors need to make sure the same amount of Xcc was used for both CTV empty vector and the RNAi. It is suggested a blink test is needed here.

We used the same amount of Xcc to inoculate CTV empty vector and the RNAi. In either inoculation, the cultured Xcc cells were suspended in sterile distilled water to a final concentration of 108 CFU/mL (OD600 = 0.3).

Point 6: Figure 1. Please draw a figure to clearly show the location of the EBE in the promoter of Cs9g12620, including the transcription start site, and translational start site.

The EBE in Cs9g12620 promoter was indicated by underlined in Figure supplement 1. We did not sure about the translation start site, but the translation start site was labelled.